# Efficient Approximation of Deep ReLU Networks for Functions on Low Dimensional Manifolds

**Minshuo Chen**    **Haoming Jiang**    **Wenjing Liao**    **Tuo Zhao**
Georgia Institute of Technology
{mchen393, jianghm, wliao60, tourzhao}@gatech.edu

## Abstract

Deep neural networks have revolutionized many real world applications, due to their flexibility in data fitting and accurate predictions for unseen data. A line of research reveals that neural networks can approximate certain classes of functions with an arbitrary accuracy, while the size of the network scales exponentially with respect to the data dimension. Empirical results, however, suggest that networks of moderate size already yield appealing performance. To explain such a gap, a common belief is that many data sets exhibit low dimensional structures, and can be modeled as samples near a low dimensional manifold. In this paper, we prove that neural networks can efficiently approximate functions supported on low dimensional manifolds. The network size scales exponentially in the approximation error, with an exponent depending on the intrinsic dimension of the data and the smoothness of the function. Our result shows that exploiting low dimensional data structures can greatly enhance the efficiency in function approximation by neural networks. We also implement a sub-network that assigns input data to their corresponding local neighborhoods, which may be of independent interest.

## 1 Introduction

In the past decade, neural networks have made astonishing breakthroughs in many real world applications, such as computer vision (Krizhevsky et al., 2012; Goodfellow et al., 2014; Long et al., 2015), natural language processing (Graves et al., 2013; Bahdanau et al., 2014; Young et al., 2018), healthcare (Miotto et al., 2017; Jiang et al., 2017), robotics (Gu et al., 2017), etc.

Although data sets in these applications are highly complex, neural networks have achieved overwhelming successes. For image classification, the winner of the 2017 ImageNet challenge retained a top-5 error rate of $2.25\%$ (Hu et al., 2018), while the data set consists of about 1.2 million labeled high resolution images in 1000 categories. For speech recognition, Amodei et al. (2016) reported that deep neural networks outperformed humans with a $5.15\%$ word error rate on the LibriSpeech corpus constructed from audio books (Panayotov et al., 2015). Such a data set consists of approximately 1000 hours of 16kHz read English speech from 8000 audio books. These empirical results suggest that neural networks can well approximate complex distributions and functions on data.

A line of research attempts to explain the success of neural networks through the lens of expressivity — neural networks can effectively approximate various classes of functions. Among existing works, the most well-known results are the universal approximation theorems, see Irie and Miyake (1988); Funahashi (1989); Cybenko (1989); Hornik (1991); Chui and Li (1992); Leshno et al. (1993). Specifically, Cybenko (1989) showed that neural networks with one single hidden layer and continuous sigmoidal[1] activations can approximate continuous functions in a unit cube with arbitrary accuracy. Later, Hornik (1991) extended the universal approximation theorem to general feed-forward networks

with a single hidden layer, while the width of the network has to be exponentially large. Specific approximation rates of shallow networks (with one hidden layer) with smooth activation functions were given in Barron (1993) and Mhaskar (1996). Recently, Lu et al. (2017) proved the universal approximation theorem for width-bounded deep neural networks, and Hanin (2017) improved the result with ReLU (Rectified Linear Units) activations, i.e. $\text{ReLU}(x) = \max\{0, x\}$. Yarotsky (2017) further showed that deep ReLU networks can uniformly approximate functions in Sobolev spaces, while the network size scales exponentially in the approximation error with an exponent depending on the data dimension. Moreover, the network size in Yarotsky (2017) matches its lower bound.

The network size considered in applications, however, is significantly smaller than what is predicted by the theory above. In the ImageNet challenge, data are RGB images with a resolution of $224 \times 224$. The theory above suggests that, to achieve a $\epsilon$ uniform approximation error, the number of neurons has to scale as $\epsilon^{-224 \times 224 \times 3/2}$ (Barron, 1993). Setting $\epsilon = 0.1$ already gives rise to $10^{224 \times 224 \times 3/2}$ neurons. However, the AlexNet (Krizhevsky et al., 2012) only consists of 650000 neurons and 60 million parameters to beat the state-of-the-art. To boost the performance on the ImageNet, several more sophisticated network structures were proposed later, such as VGG16 (Simonyan and Zisserman, 2014) which consists of about 138 million parameters. The size of both networks remains extremely small compared to $10^{224 \times 224 \times 3/2}$. Why is there a tremendous gap between theory and practice?

A common belief is that real world data sets often exhibit low dimensional structures. Many images consist of projections of 3-dimensional objects followed by some transformations, such as rotation, translation, and skeleton. Such a generating mechanism induces a small number of intrinsic parameters. Speech data are composed of words and sentences following the grammar, and therefore have a small degree of freedom. More broadly, visual, acoustic, textual, and many other types of data all have low dimensional structures due to rich local regularities, global symmetries, repetitive patterns, or redundant sampling. It is plausible to model these data as samples near a low dimensional manifold (Tenenbaum et al., 2000; Roweis and Saul, 2000). Then a natural question is:

*Can deep neural networks efficiently approximate functions supported on low dimensional manifolds?*

Function approximation on manifolds has been well studied using local polynomials (Bickel et al., 2007) and wavelets (Coifman and Maggioni, 2006). However, studies using neural networks are very limited. Two noticeable works are Chui and Mhaskar (2016) and Shaham et al. (2018). In Chui and Mhaskar (2016), high order differentiable functions on manifolds are approximated by neural networks with smooth activations, e.g., sigmoid activations and rectified quadratic unit functions $(\sigma(x) = (\max\{0, x\})^2)$. These smooth activations, however, are rarely used in the mainstream applications such as computer vision (Krizhevsky et al., 2012; Long et al., 2015; Hu et al., 2018). In Shaham et al. (2018), a 4-layer network with ReLU activations was proposed to approximate $C^2$ functions on low dimensional manifolds that have absolutely summable wavelet coefficients. However, this theory does not cover arbitrarily smooth functions, and the analysis is built upon a restrictive assumption — there exists a linear transformation that maps the input data to sparse coordinates, but such transformation is not explicitly given.

In this paper, we propose a framework to construct deep neural networks with nonsmooth activations to approximate functions supported on a $d$-dimensional smooth manifold isometrically embedded in $\mathbb{R}^D$. We prove that, in order to achieve a fixed approximation error, the network size scales exponentially with respect to the intrinsic dimension $d$, instead of the ambient dimension $D$. Our framework is flexible: **1)**. It applies to nonsmooth activations, e.g., ReLU and leaky ReLU activations; **2)**. It applies to a wide class of functions, such as Sobolev and Hölder classes which are typical examples in nonparametric statistics (Györfi et al., 2006); **3)**. It exploits high order smoothness of functions for making the approximation as efficient as possible.

**Theorem** (informal). Let $\mathcal{M}$ be a $d$-dimensional compact Riemannian manifold isometrically embedded in $\mathbb{R}^D$ with $d \ll D$. Assume $\mathcal{M}$ satisfies some mild regularity conditions. Given any $\epsilon \in (0, 1)$, there exists a ReLU neural network structure such that, for any $C^n$ $(n \geq 1)$ function $f : \mathcal{M} \to \mathbb{R}$, if the weight parameters are properly chosen, the network yields a function $\widehat{f}$ satisfying $\|f - \widehat{f}\|_\infty \leq \epsilon$. Such a network has no more than $c_1 \left(\log \frac{1}{\epsilon} + \log D\right)$ layers, and at most $c_2 \left(\epsilon^{-d/n} \log \frac{1}{\epsilon} + D \log \frac{1}{\epsilon} + D \log D\right)$ neurons and weight parameters, where $c_1, c_2$ depend on $d, n, f$, and $\mathcal{M}$.

Our network size scales like $\epsilon^{-d/n}$ and only weakly depends on the ambient dimension $D$. This is consistent with empirical observations, and partially justifies why the networks of moderate size have

achieved a great success on aforementioned learning tasks. Moreover, we show that our network size matches its lower bound up to a logarithmic factor (see Theorem 2).

Our theory applies to general $C^n$ functions and leverages the benefits of exploiting high order smoothness. Our result improves Shaham et al. (2018) for $C^n$ functions with $n > 2$. In this case, our network size scales like $\epsilon^{-d/n}$, which is significantly smaller than the one in Shaham et al. (2018) in the order of $\epsilon^{-d/2}$.

Here we state the theorem for $C^n$ functions for simplicity, and similar results hold for Hölder functions (see Theorem 1). Our framework can be easily applied to leaky ReLU activations, since leaky ReLU can be implemented by the difference of two ReLU functions.

The high level idea of our framework is to partition the low dimensional manifold into a collection of open sets, and then use Taylor expansions to approximate the function in each neighborhood. A new technique is developed to implement a sub-network that assigns the input to its corresponding neighborhood on the manifold, which may be of independent interest.

**Notations:** We use bold-faced letters to denote vectors, and normal font letters with a subscript to denote its coordinate, e.g., $\mathbf{x} \in \mathbb{R}^d$ and $x_k$ being the $k$-th coordinate of $\mathbf{x}$. Given a vector $\mathbf{n} = [n_1, \ldots, n_d]^\top \in \mathbb{N}^d$, we define $\mathbf{n}! = \prod_{i=1}^d n_i!$ and $|\mathbf{n}| = \sum_{i=1}^d n_i$. We define $\mathbf{x}^\mathbf{n} = \prod_{i=1}^d x_i^{n_i}$. Given a function $f : \mathbb{R}^d \mapsto \mathbb{R}$, we denote its derivative as $D^\mathbf{n} f = \frac{\partial^{|\mathbf{n}|} f}{\partial x_1^{n_1} \ldots \partial x_d^{n_d}}$, and its $\ell_\infty$ norm as $\|f\|_\infty = \max_\mathbf{x} |f(\mathbf{x})|$. We use $\circ$ to denote the composition operator.

## 2 Preliminaries

We briefly review manifolds, partition of unity, and function spaces defined on smooth manifolds. Details can be found in Tu (2010) and Lee (2006).

Let $\mathcal{M}$ be a $d$-dimensional Riemannian manifold isometrically embedded in $\mathbb{R}^D$.

**Definition 1** (Chart). A chart for $\mathcal{M}$ is a pair $(U, \phi)$ such that $U \subset \mathcal{M}$ is open and $\phi : U \mapsto \mathbb{R}^d$, where $\phi$ is a homeomorphism (i.e., bijective, $\phi$ and $\phi^{-1}$ are both continuous).

The open set $U$ is called a coordinate neighborhood, and $\phi$ is called a coordinate system on $U$. A chart essentially defines a local coordinate system on $\mathcal{M}$. We say two charts $(U, \phi)$ and $(V, \psi)$ on $\mathcal{M}$ are $C^k$ compatible if and only if the transition functions, $\phi \circ \psi^{-1} : \psi(U \cap V) \mapsto \phi(U \cap V)$ and $\psi \circ \phi^{-1} : \phi(U \cap V) \mapsto \psi(U \cap V)$ are both $C^k$. Then we give the definition of an atlas.

**Definition 2** ($C^k$ Atlas). An atlas for $\mathcal{M}$ is a collection $\{(U_\alpha, \phi_\alpha)\}_{\alpha \in \mathcal{A}}$ of pairwise $C^k$ compatible charts such that $\bigcup_{\alpha \in \mathcal{A}} U_\alpha = \mathcal{M}$.

**Definition 3** (Smooth Manifold). A smooth manifold is a manifold $\mathcal{M}$ together with a $C^\infty$ atlas.

Classical examples of smooth manifolds are the Euclidean space $\mathbb{R}^D$, the torus, and the unit sphere. The existence of an atlas on $\mathcal{M}$ allows us to define differentiable functions.

**Definition 4** ($C^n$ Functions on $\mathcal{M}$). Let $\mathcal{M}$ be a smooth manifold in $\mathbb{R}^D$. A function $f : \mathcal{M} \mapsto \mathbb{R}$ is $C^n$ if for any chart $(U, \phi)$, the composition $f \circ \phi^{-1} : \phi(U) \mapsto \mathbb{R}$ is continuously differentiable up to order $n$.

**Remark 1.** The definition of $C^n$ functions is independent of the choice of the chart $(U, \phi)$. Suppose $(V, \psi)$ is another chart and $V \bigcap U \neq \emptyset$. Then we have $f \circ \psi^{-1} = (f \circ \phi^{-1}) \circ (\phi \circ \psi^{-1})$. Since $\mathcal{M}$ is a smooth manifold, $(U, \phi)$ and $(V, \psi)$ are $C^\infty$ compatible. Thus, $f \circ \phi^{-1}$ is $C^n$ and $\phi \circ \psi^{-1}$ is $C^\infty$, and their composition is $C^n$.

We next introduce partition of unity, which plays a crucial role in our construction of neural networks.

**Definition 5** (Partition of Unity). A $C^\infty$ partition of unity on a manifold $\mathcal{M}$ is a collection of nonnegative $C^\infty$ functions $\rho_\alpha : \mathcal{M} \mapsto \mathbb{R}_+$ for $\alpha \in \mathcal{A}$ such that **1)**. the collection of supports, $\{\text{supp}(\rho_\alpha)\}_{\alpha \in \mathcal{A}}$ is locally finite[2]; **2)**. $\sum \rho_\alpha = 1$.

For a smooth manifold, a $C^\infty$ partition of unity always exisits.

**Proposition 1** (Existence of a $C^\infty$ partition of unity). Let $\{U_\alpha\}_{\alpha \in \mathcal{A}}$ be an open cover of a smooth manifold $\mathcal{M}$. Then there is a $C^\infty$ partition of unity $\{\rho_i\}_{i=1}^\infty$ with every $\rho_i$ having a compact support such that $\mathrm{supp}(\rho_i) \subset U_\alpha$ for some $\alpha \in \mathcal{A}$.

Proposition 1 gives rise to the decomposition $f = \sum_{i=1}^\infty f_i$ with $f_i = f\rho_i$. Note that the $f_i$'s have the same regularity as $f$, since $f_i \circ \phi^{-1} = (f \circ \phi^{-1}) \times (\rho_i \circ \phi^{-1})$ for a chart $(U, \phi)$. This decomposition has the advantage that every $f_i$ is only supported in a single chart. Then the approximation of $f$ boils down to the approximations of the $f_i$'s, which are localized and have the same regularity as $f$.

To characterize the curvature of a manifold, we adopt the following geometric concept.

**Definition 6** (Reach, *Definition 2.1* in Aamari et al. (2019)). Denote $\mathcal{C}(\mathcal{M}) = \{\mathbf{x} \in \mathbb{R}^D : \exists \mathbf{p} \neq \mathbf{q} \in \mathcal{M}, \|\mathbf{p} - \mathbf{x}\|_2 = \|\mathbf{q} - \mathbf{x}\|_2 = \inf_{\mathbf{y} \in \mathcal{M}} \|\mathbf{y} - \mathbf{x}\|_2\}$ as the set of points that have at least two nearest neighbors on $\mathcal{M}$. Then the reach $\tau > 0$ is defined as $\tau = \inf_{\mathbf{x} \in \mathcal{M}, \mathbf{y} \in \mathcal{C}(\mathcal{M})} \|\mathbf{x} - \mathbf{y}\|_2$.

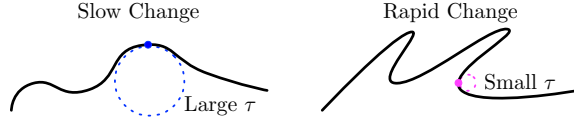

Figure 1: Manifolds with large and small reach.

Reach has a straightforward geometrical interpretation: At each point $\mathbf{x} \in \mathcal{M}$, the radius of the osculating circle is greater or equal to $\tau$. A large reach for $\mathcal{M}$ essentially requires the manifold $\mathcal{M}$ not to change "rapidly" as shown in Figure 1.

Reach determines a proper choice of an atlas for $\mathcal{M}$. In Section 4, we choose each chart $U_\alpha$ contained in a ball of radius less than $\tau/2$. For smooth manifolds with a small $\tau$, we need a large number of charts. Therefore, the reach of a smooth manifold reflects the difficulty of function approximations on $\mathcal{M}$.

## 3 Main Result

We next present how to construct a ReLU network to approximate $f : \mathcal{M} \mapsto \mathbb{R}$ with error $\epsilon$, under certain assumptions on $\mathcal{M}$ and $f$.

**Assumption 1.** $\mathcal{M}$ is a $d$-dimensional compact Riemannian manifold isometrically embedded in $\mathbb{R}^D$. There exists a constant $B$ such that for any point $\mathbf{x} \in \mathcal{M}$, we have $|x_i| \leq B$ for all $i = 1, \ldots, D$.

**Assumption 2.** The reach of $\mathcal{M}$ is $\tau > 0$.

**Assumption 3.** $f : \mathcal{M} \mapsto \mathbb{R}$ belongs to the Hölder space $H^{n,\alpha}$ with a positive integer $n$ and $\alpha \in (0, 1]$, in the sense that $f \in C^{n-1}$ and for any chart $(U, \phi)$ and $|\mathbf{n}| = n$, we have

$$\left| D^{\mathbf{n}}(f \circ \phi^{-1})\big|_{\phi(\mathbf{x}_1)} - D^{\mathbf{n}}(f \circ \phi^{-1})\big|_{\phi(\mathbf{x}_2)} \right| \leq \|\phi(\mathbf{x}_1) - \phi(\mathbf{x}_2)\|_2^\alpha, \quad \forall \mathbf{x}_1, \mathbf{x}_2 \in U. \qquad (1)$$

Assumption 3 says that all $n$-th order derivatives of $f \circ \phi^{-1}$ are Hölder continuous. Here Hölder functions are defined on manifolds. We recover the standard Hölder class on Euclidean spaces by taking $\phi$ as the identity map. We also note that Assumption 3 does not depend on the choice of charts.

We now formally state our main result. Extensions to functions in Sobolev spaces are straightforward.

**Theorem 1.** Suppose Assumptions 1 and 2 hold. Given any $\epsilon \in (0, 1)$, there exists a ReLU network structure such that, for any $f : \mathcal{M} \to \mathbb{R}$ satisfying Assumption 3, if the weight parameters are properly chosen, the network yields a function $\widehat{f}$ satisfying $\|\widehat{f} - f\|_\infty \leq \epsilon$. Such a network has no more than $c_1(\log \frac{1}{\epsilon} + \log D)$ layers, and at most $c_2(\epsilon^{-\frac{d}{n+\alpha}} \log \frac{1}{\epsilon} + D \log \frac{1}{\epsilon} + D \log D)$ neurons and weight parameters, where $c_1, c_2$ depend on $d, n, f, \tau$, and the surface area of $\mathcal{M}$.

The network structure identified by Theorem 1 consists of three sub-networks as shown in Figure 2:

- *Chart determination sub-network*, which assigns the input to its corresponding neighborhoods;

- *Taylor approximation sub-network*, which approximates $f$ by polynomials in each neighborhood;

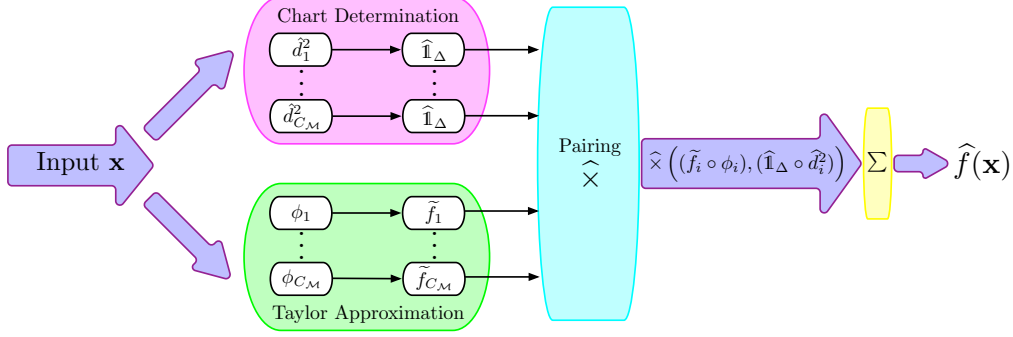

Figure 2: The ReLU network identified by Theorem 1.

- *Pairing sub-network*, which yields multiplications of the proper pairs of outputs from the chart determination and the Taylor approximation sub-networks.

Specifically, we partition the manifold as $\mathcal{M} = \bigcup_{i=1}^{C_\mathcal{M}} U_i$, where the $U_i$'s are open sets contained in a Euclidean ball of radius less than $\tau/2$. $C_\mathcal{M}$ depends on the reach $\tau$, the surface area of $\mathcal{M}$, and the dimension $d$ (see Section 4 for an explicit characterization). For each chart, the chart determination sub-network computes an approximation of the indicator function on $U_i$. The Taylor approximation sub-network provides a local polynomial approximation of $f$ on $U_i$. Then the pairing sub-network approximates the product for the proper pairs of outputs in the previous two sub-networks. Finally, $\widehat{f}$ is obtained by taking a sum over $C_\mathcal{M}$ outputs from the pairing sub-network.

The size of our ReLU network matches its lower bound up to a logarithmic factor for the approximation of functions in Hölder spaces. Denote $F^{n,d}$ as functions defined on $[0,1]^d$ in the Hölder space $H^{n-1,1}$. We state a lower bound due to DeVore et al. (1989).

**Theorem 2.** Fix $d$ and $n$. Let $W$ be a positive integer and $\kappa : \mathbb{R}^W \mapsto C([0,1]^d)$ be any mapping. Suppose there is a continuous map $\Theta : F^{n,d} \mapsto \mathbb{R}^W$ such that $\|f - \kappa(\Theta(f))\|_\infty \leq \epsilon$ for any $f \in F^{n,d}$. Then $W \geq c\epsilon^{-\frac{d}{n}}$ with $c$ depending on $n$ only.

We take $\mathbb{R}^W$ as the parameter space of a ReLU network, and $\kappa$ as the network structure. Then to approximate any $f \in F^{n,d}$, the ReLU network has at least $c\epsilon^{-\frac{d}{n}}$ weight parameters. Although Theorem 2 holds for functions on $[0,1]^d$, our network size remains in the same order up to a logarithmic factor even when the function is supported on a manifold of dimension $d$.

## 4 Proof of the Main Result

Due to limited space, we present a sketch of the proof for Theorem 1. Before we proceed, we show how to approximate the multiplication operation using ReLU networks. This operation is heavily used in the Taylor approximation sub-network, since Taylor polynomials involve sum of products. We first show ReLU networks can approximate quadratic functions.

**Lemma 1** (Proposition 2 in Yarotsky (2017)). The function $f(x) = x^2$ with $x \in [0,1]$ can be approximated by a ReLU network with any error $\epsilon > 0$. The network has depth and the number of neurons and weight parameters no more than $c\log(1/\epsilon)$ with an absolute constant $c$.

This lemma is proved in Appendix A.1. The idea is to approximate quadratic functions using a weighted sum of a series of sawtooth functions. Those sawtooth functions are obtained by compositing the triangular function

$$g(x) = 2\text{ReLU}(x) - 4\text{ReLU}(x - 1/2) + 2\text{ReLU}(x - 1),$$

which can be implemented by a single layer ReLU network.

We then approximate the multiplication operation by invoking the identity $ab = \frac{1}{4}((a+b)^2 - (a-b)^2)$ where the two squares can be approximated by ReLU networks in Lemma 1.

**Corollary 1** (Proposition 3 in Yarotsky (2017)). Given a constant $C > 0$ and $\epsilon \in (0, C^2)$, there is a ReLU network which implements a function $\widehat{\times} : \mathbb{R}^2 \mapsto \mathbb{R}$ such that: **1)**. For all inputs $x$ and

$y$ satisfying $|x| \leq C$ and $|y| \leq C$, we have $|\widehat{\times}(x,y) - xy| \leq \epsilon$; **2)**. The depth and the weight parameters of the network is no more than $c \log \frac{C^2}{\epsilon}$ with an absolute constant $c$.

The ReLU network in Theorem 1 is constructed in the following 5 steps.

**Step 1. Construction of an atlas**. Denote the open Euclidean ball with center $\mathbf{c}$ and radius $r$ in $\mathbb{R}^D$ by $\mathcal{B}(\mathbf{c}, r)$. For any $r$, the collection $\{\mathcal{B}(\mathbf{x}, r)\}_{\mathbf{x} \in \mathcal{M}}$ is an open cover of $\mathcal{M}$. Since $\mathcal{M}$ is compact, there exists a finite collection of points $\mathbf{c}_i$ for $i = 1, \dots, C_{\mathcal{M}}$ such that $\mathcal{M} \subset \bigcup_i \mathcal{B}(\mathbf{c}_i, r)$.

Now we pick the radius $r < \tau/2$ so that $U_i = \mathcal{M} \cap \mathcal{B}(\mathbf{c}_i, r)$ is diffeomorphic[3] to a ball in $\mathbb{R}^d$ (Niyogi et al., 2008). Let $\{(U_i, \phi_i)\}_{i=1}^{C_{\mathcal{M}}}$ be an atlas on $\mathcal{M}$, where $\phi_i$ is to be defined in **Step 2**. The number of charts $C_{\mathcal{M}}$ is upper bounded by

$$C_{\mathcal{M}} \leq \left\lceil \frac{SA(\mathcal{M})}{r^d} T_d \right\rceil,$$

where $SA(M)$ is the surface area of $\mathcal{M}$, and $T_d$ is the thickness[4] of the $U_i$'s.

**Remark 2.** The thickness $T_d$ scales approximately linear in $d$. As shown in Conway et al. (1987), there exists covering with $\frac{d}{e\sqrt{e}} \lesssim T_d \leq d \log d + d \log \log d + 5d$.

**Step 2. Projection with rescaling and translation**. We denote the tangent space at $\mathbf{c}_i$ as $T_{\mathbf{c}_i}(\mathcal{M}) = \text{span}(\mathbf{v}_{i1}, \dots, \mathbf{v}_{id})$, where $\{\mathbf{v}_{i1}, \dots, \mathbf{v}_{id}\}$ form an orthonormal basis. We obtain the matrix $V_i = [\mathbf{v}_{i1}, \dots, \mathbf{v}_{id}] \in \mathbb{R}^{D \times d}$ by concatenating $\mathbf{v}_{ij}$'s as column vectors.

Define $\phi_i(\mathbf{x}) = b_i(V_i^\top(\mathbf{x} - \mathbf{c}_i) + \mathbf{s}_i) \in [0,1]^d$ for any $\mathbf{x} \in U_i$, where $b_i \in (0, 1]$ is a scaling factor and $\mathbf{s}_i$ is a translation vector. Since $U_i$ is bounded, we can choose proper $b_i$ and $\mathbf{s}_i$ to guarantee $\phi_i(\mathbf{x}) \in [0,1]^d$. We rescale and translate the projection to ease the notation for the development of local Taylor approximations in **Step 4**. We also remark that each $\phi_i$ is a linear function, and can be realized by a single-layer linear network.

**Step 3. Chart determination**. This step is to locate the charts that a given input $\mathbf{x}$ belongs to. This avoids projecting $\mathbf{x}$ using unmatched charts (i.e., $\mathbf{x} \notin U_j$ for some $j$) as illustrated in Figure 3.

Proper charts[5] can be determined by compositing an indicator function and the squared Euclidean distance $d_i^2(\mathbf{x}) = \|\mathbf{x} - \mathbf{c}_i\|_2^2 = \sum_{j=1}^D (x_j - c_{i,j})^2$ for $i = 1, \dots, C_{\mathcal{M}}$. The squared distance $d_i^2(\mathbf{x})$ is a sum of univariate quadratic functions, thus, we can apply Lemma 1 to approximate $d_i^2(\mathbf{x})$ by ReLU networks. Denote $\widehat{h}_{\text{sq}}$ as an approximation of the quadratic function $x^2$ on $[0,1]$ with an approximation error $\nu$. Then we define

$$\widehat{d}_i^2(\mathbf{x}) = 4B^2 \sum_{j=1}^D \widehat{h}_{\text{sq}}\left( \left| \frac{x_j - c_{i,j}}{2B} \right| \right).$$

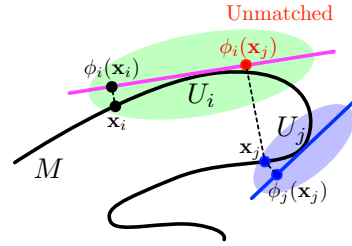

Figure 3: Projecting $\mathbf{x}_j$ using a matched chart (blue) $(U_j, \phi_j)$, and an unmatched chart (green) $(U_i, \phi_i)$.

as an approximation of $d_i^2(\mathbf{x})$. The approximation error is $\|\widehat{d}_i^2 - d_i^2\|_\infty \leq 4B^2 D\nu$, by the triangle inequality. We next consider an approximation of the indicator function of an interval as in Figure 4:

$$\widehat{\mathbb{1}}_\Delta(a) = \begin{cases} 1 & a \leq r^2 - \Delta + 4B^2 m\nu \\ -\frac{1}{\Delta - 8B^2 m\nu} a + \frac{r^2 - 4B^2 m\nu}{\Delta - 8B^2 m\nu} & a \in [r^2 - \Delta + 4B^2 m\nu, r^2 - 4B^2 m\nu], \\ 0 & a > r^2 - 4B^2 m\nu \end{cases} \quad (2)$$

where $\Delta$ ($\Delta \geq 8B^2 m\nu$) will be chosen later according to the accuracy $\epsilon$. Note that $\widehat{\mathbb{1}}_\Delta$ can be implemented exactly by a single layer ReLU network: $\widehat{\mathbb{1}}_\Delta(a) = \frac{1}{\Delta - 8B^2 m\nu} \text{ReLU}(-a + r^2 -$

$4B^2m\nu) - \frac{1}{\Delta - 8B^2m\nu}\mathrm{ReLU}(-a + r^2 - \Delta + 4B^2m\nu)$. We use $\widehat{\mathbb{1}}_\Delta \circ \widehat{d_i^2}$ to approximate the indicator function on $U_i$: if $\mathbf{x} \notin U_i$, i.e., $d_i^2(\mathbf{x}) \geq r^2$, we have $\widehat{\mathbb{1}}_\Delta \circ \widehat{d_i^2}(\mathbf{x}) = 0$; if $\mathbf{x} \in U_i$ and $d_i^2(\mathbf{x}) \leq r^2 - \Delta$, we have $\widehat{\mathbb{1}}_\Delta \circ \widehat{d_i^2}(\mathbf{x}) = 1$.

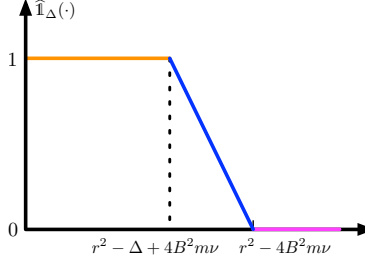

Figure 4: The approximation of the indicator function $\widehat{\mathbb{1}}_\Delta$ in (2).

**Step 4. Taylor approximation.** In each chart $(U_i, \phi_i)$, we locally approximate $f$ using Taylor polynomials of order $n$. Specifically, we decompose $f$ as $f = \sum_{i=1}^{C_\mathcal{M}} f_i$ with $f_i = f\rho_i$ where $\rho_i$ is an element in a $C^\infty$ partition of unity on $\mathcal{M}$ which is supported inside $U_i$. The existence of such a partition of unity is guaranteed by Proposition 1. Since $\mathcal{M}$ is compact and $\rho_i$ is $C^\infty$, $f_i$ preserves the regularity (smoothness) of $f$ such that $f_i \in H^{n,\alpha}$ for $i = 1, \ldots, C_\mathcal{M}$.

**Lemma 2.** Suppose Assumption 3 holds. For $i = 1, \ldots, C_\mathcal{M}$, the function $f_i$ belongs to $H^{n,\alpha}$: there exists a Hölder coefficient $L_i$ depending on $d$, $f_i$, and $\phi_i$ such that for any $|\mathbf{n}| = n$, we have

$$\left| D^\mathbf{n}(f_i \circ \phi_i^{-1})\big|_{\phi_i(\mathbf{x_1})} - D^\mathbf{n}(f_i \circ \phi_i^{-1})\big|_{\phi_i(\mathbf{x_2})} \right| \leq L_i \left\| \phi_i(\mathbf{x_1}) - \phi_i(\mathbf{x_2}) \right\|_2^\alpha, \quad \forall \mathbf{x_1}, \mathbf{x_2} \in U_i.$$

*Proof Sketch.* We provide a sketch here. Details can be found in Appendix B.1. Denote $g_1 = f \circ \phi_i^{-1}$ and $g_2 = \rho_i \circ \phi_i^{-1}$. We have $D^\mathbf{n}(f_i \circ \phi_i^{-1}) = D^\mathbf{n}(g_1 \times g_2) = \sum_{|\mathbf{p}|+|\mathbf{q}|=n} \binom{n}{|\mathbf{p}|} D^\mathbf{p} g_1 D^\mathbf{q} g_2$, by the Leibniz rule. Consider each term in the sum: for any $\mathbf{x_1}, \mathbf{x_2} \in U_i$,

$$\left| D^\mathbf{p} g_1 D^\mathbf{q} g_2 \big|_{\phi_i(\mathbf{x_1})} - D^\mathbf{p} g_1 D^\mathbf{q} g_2 \big|_{\phi_i(\mathbf{x_2})} \right|$$
$$\leq |D^\mathbf{p} g_1(\phi_i(\mathbf{x_1}))| \left| D^\mathbf{q} g_2 \big|_{\phi_i(\mathbf{x_1})} - D^\mathbf{q} g_2 \big|_{\phi_i(\mathbf{x_2})} \right| + |D^\mathbf{q} g_2(\phi_i(\mathbf{x_2}))| \left| D^\mathbf{p} g_1 \big|_{\phi_i(\mathbf{x_1})} - D^\mathbf{p} g_1 \big|_{\phi_i(\mathbf{x_2})} \right|$$
$$\leq \lambda_i \theta_{i,\alpha} \left\| \phi_i(\mathbf{x_1}) - \phi_i(\mathbf{x_2}) \right\|_2^\alpha + \mu_i \beta_{i,\alpha} \left\| \phi_i(\mathbf{x_1}) - \phi_i(\mathbf{x_2}) \right\|_2^\alpha.$$

Here $\lambda_i$ and $\mu_i$ are uniform upper bounds on the derivatives of $g_1$ and $g_2$ with order up to $n$, respectively. The last inequality above is derived as follows: by the mean value theorem, we have

$$\left| D^\mathbf{q} g_2 \big|_{\phi_i(\mathbf{x_1})} - D^\mathbf{q} g_2 \big|_{\phi_i(\mathbf{x_2})} \right| \leq \sqrt{d}\mu_i \left\| \phi_i(\mathbf{x_1}) - \phi_i(\mathbf{x_2}) \right\|_2$$
$$= \sqrt{d}\mu_i \left\| \phi_i(\mathbf{x_1}) - \phi_i(\mathbf{x_2}) \right\|_2^{1-\alpha} \left\| \phi_i(\mathbf{x_1}) - \phi_i(\mathbf{x_2}) \right\|_2^\alpha \leq \sqrt{d}\mu_i (2r)^{1-\alpha} \left\| \phi_i(\mathbf{x_1}) - \phi_i(\mathbf{x_2}) \right\|_2^\alpha,$$

where the last inequality is due to the fact that $\left\| \phi_i(\mathbf{x_1}) - \phi_i(\mathbf{x_2}) \right\|_2 \leq b_i \left\| V_i \right\| \left\| \mathbf{x_1} - \mathbf{x_2} \right\|_2 \leq 2r$. Then we set $\theta_{i,\alpha} = \sqrt{d}\mu_i(2r)^{1-\alpha}$ and by a similar argument, we set $\beta_{i,\alpha} = \sqrt{d}\lambda_i(2r)^{1-\alpha}$. We complete the proof by taking $L_i = 2^{n+1}\sqrt{d}\lambda_i\mu_i(2r)^{1-\alpha}$. $\qquad\square$

Lemma 2 is crucial for the error estimation in the local approximation of $f_i \circ \phi_i^{-1}$ by Taylor polynomials. This error estimate is given in the following theorem, where some of the proof techniques are from Theorem 1 in Yarotsky (2017).

**Theorem 3.** Let $f_i = f\rho_i$ as in **Step 4**. For any $\delta \in (0, 1)$, there exists a ReLU network structure that, if the weight parameters are properly chosen, the network yields an approximation of $f_i \circ \phi_i^{-1}$ uniformly with error $\delta$. Such a network has no more than $c\left(\log \frac{1}{\delta} + 1\right)$ layers, and at most $c'\delta^{-\frac{d}{n+\alpha}}\left(\log \frac{1}{\delta} + 1\right)$ neurons and weight parameters with $c, c'$ depending on $n, d, f_i \circ \phi_i^{-1}$.

*Proof Sketch.* The detailed proof is provided in Appendix B.2. The proof consists of two steps: **1)**. Approximate $f_i \circ \phi_i^{-1}$ using a weighted sum of Taylor polynomials; **2)**. Implement the weighted sum of Taylor polynomials using ReLU networks. Specifically, we set up a uniform grid and divide $[0, 1]^d$

into small cubes, and then approximate $f_i \circ \phi_i^{-1}$ by its $n$-th order Taylor polynomial in each cube. To implement such polynomials by ReLU networks, we recursively apply the multiplication $\widehat{\times}$ operator in Corollary 1, since these polynomials are sums of the products of different variables. □

**Step 5. Estimating the total error**. We have collected all the ingredients to implement the entire ReLU network to approximate $f$ on $\mathcal{M}$. Recall that the network structure consists of 3 main sub-networks as demonstrated in Figure 2. Let $\widehat{\times}$ be an approximation to the multiplication operator in the pairing sub-network with error $\eta$. Accordingly, the function given by the whole network is

$$\widehat{f} = \sum_{i=1}^{C_{\mathcal{M}}} \widehat{\times}(\widehat{f}_i, \widehat{\mathbb{1}}_\Delta \circ \widehat{d}_i^2) \quad \text{with} \quad \widehat{f}_i = \widetilde{f}_i \circ \phi_i,$$

where $\widetilde{f}_i$ is the approximation of $f_i \circ \phi_i^{-1}$ using Taylor polynomials in Theorem 3. The total error can be decomposed to three components according to the following theorem.

**Theorem 4.** For any $i = 1, \dots, C_{\mathcal{M}}$, we have $\|\widehat{f} - f\|_\infty \leq \sum_{i=1}^{C_{\mathcal{M}}}(A_{i,1} + A_{i,2} + A_{i,3})$, where

$$A_{i,1} = \left\| \widehat{\times}(\widehat{f}_i, \widehat{\mathbb{1}}_\Delta \circ \widehat{d}_i^2) - \widehat{f}_i \times (\widehat{\mathbb{1}}_\Delta \circ \widehat{d}_i^2) \right\|_\infty \leq \eta,$$

$$A_{i,2} = \left\| \widehat{f}_i \times (\widehat{\mathbb{1}}_\Delta \circ \widehat{d}_i^2) - f_i \times (\widehat{\mathbb{1}}_\Delta \circ \widehat{d}_i^2) \right\|_\infty \leq \delta,$$

$$A_{i,3} = \left\| f_i \times (\widehat{\mathbb{1}}_\Delta \circ \widehat{d}_i^2) - f_i \times \mathbb{1}(\mathbf{x} \in U_i) \right\|_\infty \leq \frac{c(\pi + 1)}{r(1 - r/\tau)}\Delta \quad \text{for some constant } c.$$

Here $\mathbb{1}(\mathbf{x} \in U_i)$ is the indicator function on $U_i$. Theorem 4 is proved in Appendix B.3. In order to achieve an $\epsilon$ total approximation error, i.e., $\|f - \widehat{f}\|_\infty \leq \epsilon$, we need to control the errors in the three sub-networks. In other words, we need to decide $\nu$ for $\widehat{d}_i^2$, $\Delta$ for $\widehat{\mathbb{1}}_\Delta$, $\delta$ for $\widetilde{f}_i$, and $\eta$ for $\widehat{\times}$. Note that $A_{i,1}$ is the error from the pairing sub-network, $A_{i,2}$ is the approximation error in the Taylor approximation sub-network, and $A_{i,3}$ is the error from the chart determination sub-network. The error bounds on $A_{i,1}, A_{i,2}$ are straightforward from the constructions of $\widehat{\times}$ and $\widehat{f}_i$. The estimate of $A_{i,3}$ involves some technical analysis since $\|\widehat{\mathbb{1}}_\Delta \circ \widehat{d}_i^2 - \mathbb{1}(\mathbf{x} \in U_i)\|_\infty = 1$. Note that $\widehat{\mathbb{1}}_\Delta \circ \widehat{d}_i^2(\mathbf{x}) - \mathbb{1}(\mathbf{x} \in U_i) = 0$ whenever $\|\mathbf{x} - \mathbf{c}_i\|_2^2 < r^2 - \Delta$ or $\|\mathbf{x} - \mathbf{c}_i\|_2^2 > r^2$, so we only need to prove that $|f_i(\mathbf{x})|$ is sufficiently small in the region $\mathcal{K}_i$ defined below.

**Lemma 3.** For any $i = 1, \dots, C_{\mathcal{M}}$, denote $\mathcal{K}_i = \{\mathbf{x} \in \mathcal{M} : r^2 - \Delta \leq \|\mathbf{x} - \mathbf{c}_i\|_2^2 \leq r^2\}$. Then there exists a constant $c$ depending on $f_i$'s and $\phi_i$'s such that

$$\max_{\mathbf{x} \in \mathcal{K}_i} |f_i(\mathbf{x})| \leq \frac{c(\pi + 1)}{r(1 - r/\tau)}\Delta.$$

*Proof Sketch.* The detailed proof is in Appendix B.4. The function $f_i \circ \phi_i^{-1}$ is defined on $\phi_i(U_i) \subset [0, 1]^d$. We extend $f_i \circ \phi_i^{-1}$ to $[0, 1]^d$ by letting $f_i \circ \phi_i^{-1}(\mathbf{x}) = 0$ for $\mathbf{x} \in [0, 1]^d \setminus \phi_i(U_i)$. It is easy to verify that such an extension preserves the regularity of $f_i \circ \phi_i^{-1}$, since $\text{supp}(f_i)$ is a compact subset of $U_i$. By the mean value theorem, for any $\mathbf{x}, \mathbf{y} \in \mathcal{K}_i$, there exists $\mathbf{z} = \beta \phi_i(\mathbf{x}) + (1 - \beta)\phi_i(\mathbf{y})$ for some $\beta \in (0, 1)$ such that

$$|f_i(\mathbf{x}) - f_i(\mathbf{y})| \leq \|\nabla f_i \circ \phi_i^{-1}(\mathbf{z})\|_2 \|\phi_i(\mathbf{x}) - \phi_i(\mathbf{y})\|_2 \leq \|\nabla f_i \circ \phi_i^{-1}(\mathbf{z})\|_2 b_i \|V_i\|_2 \|\mathbf{x} - \mathbf{y}\|_2.$$

We pick $\mathbf{y} \in \partial U_i$ (the boundary of $U_i$) so that $f_i(\mathbf{y}) = 0$. Since $f_i \in H^{n,\alpha}$ and $\mathcal{M}$ is compact, $\left\| \nabla f_i \circ \phi_i^{-1}(\mathbf{z}) \right\|_2 b_i \|V_i\|_2 \leq c$ for some $c > 0$. To bound $|f_i(\mathbf{x})|$, the key is to estimate $\|\mathbf{x} - \mathbf{y}\|_2$. We next prove that, for any $\mathbf{x} \in \mathcal{K}_i$, there exists $\mathbf{y} \in \partial U_i$ satisfying $\|\mathbf{x} - \mathbf{y}\|_2 \leq \frac{\pi+1}{r(1-r/\tau)}\Delta$.

The idea is to consider a geodesic[6] $\gamma(t)$ parameterized by the arc length from $\mathbf{x}$ to $\partial U_i$ in Figure 5. Denote $\mathbf{y} = \partial U_i \bigcap \gamma$. Without loss of generality, we shift the center $\mathbf{c}_i$ to $\mathbf{0}$ in the following analysis. To utilize polar coordinates, we define two auxiliary quantities: $\theta(t) = \gamma(t)^\top \dot{\gamma}(t)/\|\gamma(t)\|_2$ and $\ell(t) = \|\gamma(t)\|_2$, where $\dot{\gamma}$ denotes the derivative of $\gamma$.

We show that there exists a geodesic $\gamma(t)$ satisfying $\inf_t \dot{\ell}(t) \geq \frac{1-r/\tau}{\pi+1} > 0$. This implies that the geodesic continuously moves away from the center. Denote $T$ such that $\gamma(T) = \mathbf{y}$. By the definition of geodesic, $T$ is the arc length of $\gamma(t)$ between $\mathbf{x}$ and $\mathbf{y}$. We have $T \inf_t \dot{\ell}(t) \leq \ell(T) - \ell(0) \leq r - \sqrt{r^2 - \Delta} \leq \frac{\Delta}{r}$. Therefore, $\|\mathbf{x} - \mathbf{y}\|_2 \leq T \leq \frac{\Delta}{r \inf_t \dot{\ell}(t)} \leq \frac{\pi+1}{r(1-r/\tau)}\Delta$. $\qquad\square$

Given Theorem 4, we choose

$$\eta = \delta = \frac{\epsilon}{3C_{\mathcal{M}}} \text{ and } \Delta = \frac{r(1 - r/\tau)\epsilon}{3c(\pi+1)C_{\mathcal{M}}} \qquad (3)$$

so that the approximation error is bounded by $\epsilon$. Moreover, we choose $\nu = \frac{\Delta}{16B^2D}$ to guarantee $\Delta > 8B^2 D\nu$ so that the definition of $\widehat{\mathbb{1}}_\Delta$ is valid.

Finally we quantify the size of the ReLU network. Recall that the chart determination sub-network has $c_1 \log \frac{1}{\nu}$ layers, the Taylor approximation sub-network has $c_2 \log \frac{1}{\delta}$ layers, and the pairing sub-network has $c_3 \log \frac{1}{\eta}$ layers. Here $c_2$ depends on $d, n, f$, and $c_1, c_3$ are absolute constants. Combining these with (3) yields the depth in Theorem 1. By a similar argument, we can obtain the number of neurons and weight parameters. A detailed analysis is given in Appendix B.5.

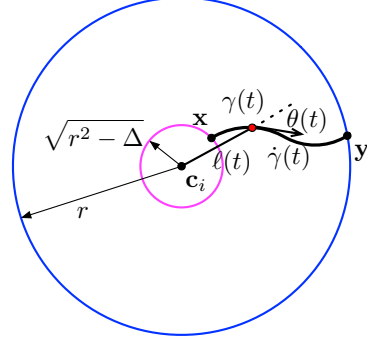

Figure 5: A geometric illustration of $\theta$ and $\ell$.

# 5 Discussions

**ReLU activations**. We consider neural networks with ReLU activations for a practical concern — ReLU activations are widely used in deep networks. Moreover, ReLU networks are easier to train compared with sigmoid or hyperbolic tangent activations, which are known for their notorious vanishing gradient problem (Goodfellow et al., 2016; Glorot et al., 2011).

**Low Dimensional Manifolds**. The low dimensional manifold model plays a vital role to reduce the network size. As shown in Theorem 2, to approximate functions in $F^{n,D}$ with accuracy $\epsilon$, the minimal number of weight parameters is $O(\epsilon^{-\frac{D}{n}})$. This lower bound is huge, and can not be improved without low dimensional structures of data.

**Existence vs. Learnability and Generalization**. Our Theorem 1 shows the existence of a ReLU network structure that gives efficient approximations of functions on low dimensional manifolds, if the weight parameters are properly chosen. In practice, it is observed that larger neural networks are easier to train and yield better generalization performances (Li et al., 2018; Zhang et al., 2016; Arora et al., 2018). This is referred to as overparameterization. Establishing the connection between learnability and generalization is an important future direction.

**Convolutional Filters**. Convolutional neural networks (CNNs, Krizhevsky et al. (2012)) are widely used in computer vision, language modeling, etc. Empirical results reveal that different convolutional filters can capture various patterns in images, e.g., edge detection filters. An interesting question is whether convolutional filters serve as charts in our framework.

**Equivalent Networks**. The ReLU network identified in Theorem 1 is sparsely connected. Several other network structures can yield the same function as our ReLU network. It is interesting to investigate whether these network structures also possess the universal approximation property.

# 6 Acknowledgements

This work is supported by NSF grants DMS 1818751 and III 1717916. The authors would like to thank Ryan Tibshirani for his helpful discussions and insightful comments.

## Footnotes

[1]A function $\sigma(x)$ is sigmoidal, if $\sigma(x) \to 0$ as $x \to -\infty$, and $\sigma(x) \to 1$ as $x \to \infty$.

[2] A collection $\{A_\alpha\}$ is locally finite if every point has a neighborhood that meets only finitely many of $A_\alpha$'s.

[3]$P$ is diffeomorphic to $Q$ if there is a mapping $\Gamma : P \mapsto Q$ bijective, $C^\infty$, and its inverse also being $C^\infty$.

[4]Thickness is the average number of $U_i$'s that contain a point on $\mathcal{M}$ (Conway et al., 1987).

[5]Note that an input $\mathbf{x}$ can belong to multiple charts. Accordingly, the chart determination sub-network determines all these charts.

[6]A geodesic is the shortest path between two points on the manifold. We refer readers to Chapter 6 in Lee (2006) for a formal introduction.

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
