[Supplementary Material · approxNN_nips_supp.pdf]

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

# Supplementary Material for Efficient Approximation of Deep Neural Networks

## A    Proofs of Preliminary Results in Section 4

### A.1    Proof of Lemma 1

*Proof.* We partition the interval $[0,1]$ uniformly into $2^N$ subintervals $I_k = [\frac{k}{2^N}, \frac{k+1}{2^N}]$ for $k = 0, \ldots, 2^N - 1$. We approximate $f(x) = x^2$ on these subintervals by a linear interpolation

$$\widehat{f}_k = \frac{2k+1}{2^N}\left(x - \frac{k}{2^N}\right) + \frac{k^2}{2^{2N}}, \quad \text{for } x \in I_k.$$

It is straightforward to check that $\widehat{f}_k$ meets $f$ at the endpoints $\frac{k}{2^N}, \frac{k+1}{2^N}$ of $I_k$.

We evaluate the approximation error of $\widehat{f}_k$ on the interval $I_k$:

$$\max_{x \in I_k}\left|f(x) - \widehat{f}_k(x)\right| = \max_{x \in I_k}\left|x^2 - \frac{2k+1}{2^N}x + \frac{k^2+k}{2^{2N}}\right|$$

$$= \max_{x \in I_k}\left|\left(x - \frac{2k+1}{2^{2N}}\right)^2 - \frac{1}{2^{4N}}\right|$$

$$= \frac{1}{2^{4N}}.$$

Note that this approximation error does not depend on $k$. Thus, in order to achieve an $\epsilon$ approximation error, we only need

$$\frac{1}{2^{4N}} \leq \epsilon \implies N \geq \frac{\log\frac{1}{\epsilon}}{4}.$$

Let $N = \lceil\frac{\log\frac{1}{\epsilon}}{4}\rceil$ and denote $f_N = \sum_{k=0}^{2^N-1}\widehat{f}_k \mathbb{1}\{x \in I_k\}$. We compute the increment from $f_{N-1}$ to $f_N$ for $x \in \left[\frac{k}{2^{N-1}}, \frac{k+1}{2^{N-1}}\right]$ as follows,

$$f_{N-1} - f_N = \begin{cases} \frac{k^2}{2^{2(N-1)}} + \frac{2k+1}{2^{N-1}}\left(x - \frac{k}{2^{N-1}}\right) - \frac{k^2}{2^{2(N-1)}} - \frac{4k+1}{2^N}\left(x - \frac{k}{2^{N-1}}\right), & x \in \left[\frac{k}{2^{N-1}}, \frac{2k+1}{2^N}\right) \\ \frac{k^2}{2^{2(N-1)}} + \frac{2k+1}{2^{N-1}}\left(x - \frac{k}{2^{N-1}}\right) - \frac{(2k+1)^2}{2^{2N}} - \frac{4k+3}{2^N}\left(x - \frac{2k+1}{2^N}\right), & x \in \left[\frac{2k+1}{2^N}, \frac{k+1}{2^{N-1}}\right) \end{cases}$$

$$= \begin{cases} \frac{1}{2^N}x - \frac{k}{2^{2N-1}}, & x \in \left[\frac{k}{2^{N-1}}, \frac{2k+1}{2^N}\right) \\ -\frac{1}{2^N}x + \frac{k+1}{2^{2N-1}}, & x \in \left[\frac{2k+1}{2^N}, \frac{k+1}{2^{N-1}}\right) \end{cases}.$$

We observe that $f_{N-1} - f_N$ is a triangular function on $\left[\frac{k}{2^{N-1}}, \frac{k+1}{2^{N-1}}\right]$. The maximum is $\frac{1}{2^{2N}}$ independent of $k$ attained at $x = \frac{2k+1}{2^N}$. The minimum is $0$ attained at the endpoints $\frac{k}{2^{N-1}}, \frac{k+1}{2^{N-1}}$. To implement $f_N$, we consider a triangular function representable by a one-layer ReLU network:

$$g(x) = 2\sigma(x) - 4\sigma(x - 0.5) + 2\sigma(x - 1).$$

Denote by $g_m = g \circ g \circ \cdots \circ g$ the composition of totally $m$ functions $g$. Observe that $g_m$ is a sawtooth function with $2^{m-1}$ peaks at $\frac{2k+1}{2^m}$ for $k = 0, \ldots, 2^{m-1} - 1$, and we have $g_m\left(\frac{2k+1}{2^m}\right) = 1$ for $k = 0, \ldots, 2^{m-1} - 1$. Then we have $f_{N-1} - f_N = \frac{1}{2^{2N}}g_N$. By induction, we have

$$f_N = f_{N-1} - \frac{1}{2^{2N}}g_N$$

$$= f_{N-2} - \frac{1}{2^{2N}}g_N - \frac{1}{2^{2N-2}}g_{N-1}$$

$$= \cdots$$

$$= x - \sum_{k=1}^{N}\frac{1}{2^{2k}}g_k.$$

Therefore, $f_N$ can be implemented by a ReLU network of depth $\lceil \frac{\log \frac{1}{\epsilon}}{4} \rceil \leq c \log \frac{1}{\epsilon}$ for an absolute constant $c$. Each layer consists of at most 3 neurons, hence, the total number of neurons and weight parameters is no more than $c' \log \frac{1}{\epsilon}$. $\qquad\square$

## A.2 Proof of Corollary 1

*Proof.* Let $\widehat{f}_\delta$ be an approximation of the quadratic function on $[0,1]$ with error $\delta \in (0,1)$. We set

$$\widehat{\times}(x,y) = C^2 \left( \widehat{f}_\delta \left( \frac{|x+y|}{2C} \right) - \widehat{f}_\delta \left( \frac{|x-y|}{2C} \right) \right).$$

Now we determine $\delta$. We bound the error of $\widehat{\times}$

$$
\begin{aligned}
|\widehat{\times}(x,y) - xy| &= C^2 \left| \widehat{f}_\delta \left( \frac{|x+y|}{2C} \right) - \frac{|x+y|^2}{4C^2} - \widehat{f}_\delta \left( \frac{|x-y|}{2C} \right) + \frac{|x-y|^2}{4C^2} \right| \\
&\leq C^2 \left| \widehat{f}_\delta \left( \frac{|x+y|}{2C} \right) - \frac{|x+y|^2}{4C^2} \right| + \left| \widehat{f}_\delta \left( \frac{|x-y|}{2C} \right) - \frac{|x-y|^2}{4C^2} \right| \\
&\leq 2C^2 \delta.
\end{aligned}
$$

Thus, we pick $\delta = \frac{\epsilon}{2C^2}$ to ensure $\left| \widehat{\times}(x,y) - xy \right| \leq \epsilon$ for any inputs $x$ and $y$. As shown in Lemma 1, we can implement $\widehat{f}_\delta$ using a ReLU network of depth at most $c' \log \frac{1}{\delta} = c \log \frac{C^2}{\epsilon}$ with absolute constants $c', c$. The proof is complete. $\qquad\square$

# B Proofs of Construction of Neural Networks in Section 4

## B.1 Proof of Lemma 2

*Proof.* We rewrite $f_i \circ \phi_i^{-1}$ as

$$\underbrace{(f \circ \phi_i^{-1})}_{g_1} \times \underbrace{(\rho_i \circ \phi_i^{-1})}_{g_2}. \tag{4}$$

By the definition of the partition of unity, we know $g_2$ is $C^\infty$. This implies that $g_2$ is $(n+1)$ continuously differentiable. Since $\operatorname{supp}(\rho_i)$ is compact, the $k$-th derivative of $g_2$ is uniformly bounded by $\lambda_{i,k}$ for any $k \leq n+1$. Let $\lambda_i = \max_{k \leq n+1} \lambda_{i,k}$. We have for any $|\mathbf{n}| \leq n$ and $\mathbf{x}_1, \mathbf{x}_2 \in U_i$,

$$
\begin{aligned}
|D^{\mathbf{n}} g_2(\phi_i(\mathbf{x}_1)) - D^{\mathbf{n}} g_2(\phi_i(\mathbf{x}_2))| &\leq \sqrt{d} \lambda_i \|\phi_i(\mathbf{x}_1) - \phi_i(\mathbf{x}_2)\|_2 \\
&\leq \sqrt{d} \lambda_i b_i^{1-\alpha} \|\mathbf{x}_1 - \mathbf{x}_2\|_2^{1-\alpha} \|\phi_i(\mathbf{x}_1) - \phi_i(\mathbf{x}_2)\|_2^\alpha.
\end{aligned}
$$

The last inequality follows from $\phi_i(\mathbf{x}) = b_i(V_i^\top(\mathbf{x} - \mathbf{c}_i) + \mathbf{s}_i)$ and $\|V_i\|_2 = 1$. Observe that $U_i$ is bounded, hence, we have $\|\mathbf{x}_1 - \mathbf{x}_2\|_2^{1-\alpha} \leq (2r)^{1-\alpha}$. Absorbing $\|\mathbf{x}_1 - \mathbf{x}_2\|_2^{1-\alpha}$ into $\sqrt{d} \lambda_i b_i^{1-\alpha}$, we have the derivative of $g_2$ is Hölder continuous. We denote $\beta_{i,\alpha} = \sqrt{d} \lambda_i b_i^{1-\alpha} (2r)^{1-\alpha} \leq \sqrt{d} \lambda_i (2r)^{1-\alpha}$. Similarly, $g_1$ is $C^{n-1}$ by Assumption 3. Then there exists a constant $\mu_i$ such that the $k$-th derivative of $g_1$ is uniformly bounded by $\mu_i$ for any $k \leq n-1$. These derivatives are also Hölder continuous with coefficient $\theta_{i,\alpha} \leq \sqrt{d} \mu_i (2r)^{1-\alpha}$.

By the Leibniz rule, for any $|\mathbf{n}| = n$, we expand the $n$-th derivative of $f_i \circ \phi_i^{-1}$ as

$$D^{\mathbf{n}}(g_1 \times g_2) = \sum_{|\mathbf{p}| + |\mathbf{q}| = n} \binom{n}{|\mathbf{p}|} D^{\mathbf{p}} g_1 D^{\mathbf{q}} g_2.$$

Consider each summand in the above right-hand side. For any $\mathbf{x}_1, \mathbf{x}_2 \in U_i$, we derive

$$
\begin{aligned}
&\left| D^{\mathbf{p}} g_1(\phi_i(\mathbf{x}_1)) D^{\mathbf{q}} g_2(\phi_i(\mathbf{x}_1)) - D^{\mathbf{p}} g_1(\phi_i(\mathbf{x}_2)) D^{\mathbf{q}} g_2(\phi_i(\mathbf{x}_2)) \right| \\
=& \left| D^{\mathbf{p}} g_1(\phi_i(\mathbf{x}_1)) D^{\mathbf{q}} g_2(\phi_i(\mathbf{x}_1)) - D^{\mathbf{p}} g_1(\phi_i(\mathbf{x}_1)) D^{\mathbf{q}} g_2(\phi_i(\mathbf{x}_2)) \right. \\
&\left. + D^{\mathbf{p}} g_1(\phi_i(\mathbf{x}_1)) D^{\mathbf{q}} g_2(\phi_i(\mathbf{x}_2)) - D^{\mathbf{p}} g_1(\phi_i(\mathbf{x}_2)) D^{\mathbf{q}} g_2(\phi_i(\mathbf{x}_2)) \right| \\
\leq& \left| D^{\mathbf{p}} g_1(\phi_i(\mathbf{x}_1)) \right| \left| D^{\mathbf{q}} g_2(\phi_i(\mathbf{x}_1)) - D^{\mathbf{q}} g_2(\phi_i(\mathbf{x}_2)) \right| \\
&+ \left| D^{\mathbf{q}} g_2(\phi_i(\mathbf{x}_2)) \right| \left| D^{\mathbf{p}} g_1(\phi_i(\mathbf{x}_1)) - D^{\mathbf{p}} g_1(\phi_i(\mathbf{x}_2)) \right| \\
\leq& \mu_i \theta_{i,\alpha} \left\| \phi_i(\mathbf{x}_1) - \phi_i(\mathbf{x}_2) \right\|_2^{\alpha} + \lambda_i \beta_{i,\alpha} \left\| \phi_i(\mathbf{x}_1) - \phi_i(\mathbf{x}_2) \right\|_2^{\alpha} \\
\leq& 2\sqrt{d} \mu_i \lambda_i (2r)^{1-\alpha} \left\| \phi_i(\mathbf{x}_1) - \phi_i(\mathbf{x}_2) \right\|_2^{\alpha} .
\end{aligned}
$$

Observe that there are totally $2^n$ summands in the right hand side of (4). Therefore, for any $\mathbf{x}_1, \mathbf{x}_2 \in U_i$ and $|\mathbf{n}| = n$, we have

$$
\left| D^{\mathbf{n}}(f_i \circ \phi_i^{-1}) \right|_{\phi_i(\mathbf{x}_1)} - D^{\mathbf{n}}(f_i \circ \phi_i^{-1}) \Big|_{\phi_i(\mathbf{x}_2)} \right| \leq 2^{n+1} \sqrt{d} \mu_i \lambda_i (2r)^{1-\alpha} \left\| \phi_i(\mathbf{x}_1) - \phi_i(\mathbf{x}_2) \right\|_2^{\alpha} .
$$

$\square$

### B.2  Proof of Theorem 3

*Proof.* The proof consists of two steps. We first approximate $f_i \circ \phi_i^{-1}$ by a Taylor polynomial, and then implement the Taylor polynomial using a ReLU network. To ease the analysis, we extend $f_i \circ \phi_i^{-1}$ to the whole cube $[0,1]^d$ by assigning $f_i \circ \phi_i^{-1}(\mathbf{x}) = 0$ for $\phi_i(\mathbf{x}) \in [0,1]^d \setminus \phi_i(U_i)$. It is straightforward to check that this extension preserves the regularity of $f_i \circ \phi_i^{-1}$, since $f_i$ vanishes on the complement of the compact set $\mathrm{supp}(\rho_i) \subset U_i$. For notational simplicity, we denote $f_i^{\phi} = f_i \circ \phi_i^{-1}$ with the extension.

**Step 1.** We define a trapezoid function

$$
\psi(x) = \begin{cases} 1 & |x| < 1 \\ 2 - |x| & 1 \leq |x| \leq 2 \\ 0 & |x| > 2 \end{cases} .
$$

Note that we have $\|\psi\|_{\infty} = 1$. Let $N$ be a positive integer, we form a uniform grid on $[0,1]^d$ by dividing each coordinate into $N$ subintervals. We then consider a partition of unity on these grid defined by

$$
\zeta_{\mathbf{m}}(\mathbf{x}) = \prod_{k=1}^{d} \psi \left( 3N \left( x_k - \frac{m_k}{N} \right) \right) .
$$

We can check that $\sum_{\mathbf{m}} \zeta_{\mathbf{m}}(\mathbf{x}) = 1$ as in Figure 6.

Figure 6: Illustration of the construction of $\zeta_{\mathbf{m}}$ on the $k$-th coordinate.

We also observe that $\mathrm{supp}(\zeta_{\mathbf{m}}) = \left\{ \mathbf{x} : \left| x_k - \frac{m_k}{N} \right| \leq \frac{1}{N}, k = 1, \ldots, d \right\}$. Now we construct a Taylor polynomial of degree $n$ for approximating $f_i^{\phi}$ at $\frac{\mathbf{m}}{N}$:

$$
P_{\mathbf{m}}(\mathbf{x}) = \sum_{|\mathbf{n}| \leq n} \frac{D^{\mathbf{n}} f_i^{\phi}}{\mathbf{n}!} \Bigg|_{\mathbf{x} = \frac{\mathbf{m}}{N}} \left( \mathbf{x} - \frac{\mathbf{m}}{N} \right)^{\mathbf{n}} .
$$

Define $\bar{f}_i = \sum_{\mathbf{m}\in\{0,\dots,N\}^d} \zeta_{\mathbf{m}} P_{\mathbf{m}}$. We bound the approximation error $\left\|\bar{f}_i - f_i^{\phi}\right\|_{\infty}$:

$$
\begin{aligned}
\max_{\mathbf{x}\in[0,1]^d}\left|\bar{f}_i(\mathbf{x}) - f_i^{\phi}(\mathbf{x})\right| &= \max_{\mathbf{x}}\left|\sum_{\mathbf{m}}\phi_{\mathbf{m}}(\mathbf{x})(P_{\mathbf{m}}(\mathbf{x}) - f_i^{\phi}(\mathbf{x}))\right| \\
&\leq \max_{\mathbf{x}}\sum_{\mathbf{m}:\left|x_k - \frac{m_k}{N}\right|\leq\frac{1}{N}}\left|P_{\mathbf{m}}(\mathbf{x}) - f_i^{\phi}(\mathbf{x})\right| \\
&\leq \max_{\mathbf{x}} 2^d \max_{\mathbf{m}:\left|x_k - \frac{m_k}{N}\right|\leq\frac{1}{N}}\left|P_{\mathbf{m}}(\mathbf{x}) - f_i^{\phi}(\mathbf{x})\right| \\
&\leq \max_{\mathbf{x}}\frac{2^d d^n}{n!}\left(\frac{1}{N}\right)^n \max_{|\mathbf{n}|=n}\left|D^{\mathbf{n}}f_i^{\phi}\big|_{\frac{\mathbf{m}}{N}} - D^{\mathbf{n}}f_i^{\phi}\big|_{\mathbf{y}}\right| \\
&\leq \max_{\mathbf{x}}\frac{2^d d^n}{n!}\left(\frac{1}{N}\right)^n 2^{n+1}\sqrt{d}\mu_i\lambda_i(2r)^{1-\alpha}\left\|\frac{\mathbf{m}}{N}-\mathbf{x}\right\|_2^{\alpha} \\
&\leq \sqrt{d}\mu_i\lambda_i(2r)^{1-\alpha}\frac{2^{d+n+1}d^{n+\alpha/2}}{n!}\left(\frac{1}{N}\right)^{n+\alpha}.
\end{aligned}
$$

Here $\mathbf{y}$ is the linear interpolation of $\frac{\mathbf{m}}{N}$ and $\mathbf{x}$, determined by the Taylor remainder. The second last inequality is obtained by the Hölder continuity in Lemma 2. By setting $\sqrt{d}\mu_i\lambda_i(2r)^{1-\alpha}\frac{2^{d+n+1}d^{n+\alpha/2}}{n!}\left(\frac{1}{N}\right)^{n+\alpha}\leq\frac{\delta}{2}$, we get $N\geq\left(\frac{\sqrt{d}\mu_i\lambda_i(2r)^{1-\alpha}2^{d+n+2}d^{n+\alpha/2}}{\delta n!}\right)^{\frac{1}{n+\alpha}}$. Accordingly, the approximation error is bounded by $\|\bar{f}_i - f_i^{\phi}\|_{\infty}\leq\frac{\delta}{2}$.

**Step 2.** We next implement $\widetilde{f}_i$ by a ReLU network that approximates $\bar{f}_i$ up to an error $\frac{\delta}{2}$. We denote

$$
P_{\mathbf{m}}(\mathbf{x}) = \sum_{|\mathbf{n}|\leq n} a_{\mathbf{m},\mathbf{n}}\left(\mathbf{x} - \frac{\mathbf{m}}{N}\right)^{\mathbf{n}},
$$

where $a_{\mathbf{m},\mathbf{n}} = \frac{D^{\mathbf{n}}f_i^{\phi}}{\mathbf{n}!}\Big|_{\mathbf{x}=\frac{\mathbf{m}}{N}}$. Then we rewrite $\bar{f}_i$ as

$$
\bar{f}_i(\mathbf{x}) = \sum_{\mathbf{m}\in\{0,\dots,N\}^d}\sum_{|\mathbf{n}|\leq n} a_{\mathbf{m},\mathbf{n}}\zeta_{\mathbf{m}}(\mathbf{x})\left(\mathbf{x} - \frac{\mathbf{m}}{N}\right)^{\mathbf{n}}. \tag{5}
$$

Note that (5) is a linear combination of products $\zeta_{\mathbf{m}}\left(\mathbf{x} - \frac{\mathbf{m}}{N}\right)^{\mathbf{n}}$. Each product involves at most $d+n$ univariate terms: $d$ terms for $\zeta_{\mathbf{m}}$ and $n$ terms for $\left(\mathbf{x} - \frac{\mathbf{m}}{N}\right)^{\mathbf{n}}$. We recursively apply Corollary 1 to implement the product. Specifically, let $\widehat{\times}_\epsilon$ be the approximation of the product operator in Corollary 1 with error $\epsilon$, which will be chosen later. Consider the following chain application of $\widehat{\times}_\epsilon$:

$$
\widetilde{f}_{\mathbf{m},\mathbf{n}}(\mathbf{x}) = \widehat{\times}_\epsilon\left(\psi(3Nx_1 - 3m_1), \widehat{\times}_\epsilon\left(\dots, \widehat{\times}_\epsilon\left(\psi(3N_dx_d - m_d), \widehat{\times}_\epsilon\left(x_1 - \frac{m_1}{N}, \dots\right)\right)\right)\right).
$$

Now we estimate the error of the above approximation. Note that we have $|\psi(3Nx_k - 3m_k)|\leq 1$ and $\left|x_k - \frac{m_k}{N}\right|\leq 1$ for all $k\in\{1,\dots,d\}$ and $\mathbf{x}\in[0,1]^d$. We then have

$$
\begin{aligned}
\left|\widetilde{f}_{\mathbf{m},\mathbf{n}}(\mathbf{x}) - \zeta_{\mathbf{m}}\left(\mathbf{x} - \frac{\mathbf{m}}{N}\right)^{\mathbf{n}}\right| &= \left|\widehat{\times}_\epsilon\left(\psi(3Nx_1 - 3m_1), \widehat{\times}_\epsilon\left(\dots, \widehat{\times}_\epsilon\left(x_1 - \frac{m_1}{N}, \dots\right)\right)\right) - \zeta_{\mathbf{m}}\left(\mathbf{x} - \frac{\mathbf{m}}{N}\right)^{\mathbf{n}}\right| \\
&\leq \Big|\widehat{\times}_\epsilon\left(\psi(3Nx_1 - 3m_1), \widehat{\times}_\epsilon(\psi(3Nx_2 - 3m_2), \dots)\right) \\
&\qquad - \psi(3N_1 - 3m_1)\widehat{\times}_\epsilon(\psi(3Nx_2 - 3m_2), \dots)\Big| \\
&\quad + \left|\psi(3Nx_1 - m_1)\right|\left|\widehat{\times}_\epsilon(\psi(3Nx_2 - 3m_2), \dots) - \psi(3Nx_2 - 3m_2)\widehat{\times}_\epsilon(\dots)\right| \\
&\quad + \dots \\
&\leq (d+n)\delta.
\end{aligned}
$$

Moreover, we have $\widetilde{f}_{\mathbf{m},\mathbf{n}}(\mathbf{x}) = \zeta_{\mathbf{m}}\left(\mathbf{x} - \frac{\mathbf{m}}{N}\right)^{\mathbf{n}} = 0$, if $\mathbf{x}\notin\mathrm{supp}(\zeta_{\mathbf{m}})$. Now we define

$$
\widetilde{f}_i = \sum_{\mathbf{m}\in\{0,\dots,N\}^d}\sum_{|\mathbf{n}|\leq n} a_{\mathbf{m},\mathbf{n}}\widetilde{f}_{\mathbf{m},\mathbf{n}}.
$$

The approximation error is bounded by

$$\max_{\mathbf{x}} \left| \widetilde{f}_i(\mathbf{x}) - \bar{f}_i(\mathbf{x}) \right| = \left| \sum_{\mathbf{m} \in \{0,\ldots,N\}^d} \sum_{|\mathbf{n}| \leq n} a_{\mathbf{m},\mathbf{n}} \left( \widetilde{f}_{\mathbf{m},\mathbf{n}}(\mathbf{x}) - \zeta_{\mathbf{m}} \left( \mathbf{x} - \frac{\mathbf{m}}{N} \right)^{\mathbf{n}} \right) \right|$$

$$\leq \max_{\mathbf{x}} \lambda_i \mu_i 2^{d+n+1} \max_{\mathbf{m}: \mathbf{x} \in \mathrm{supp}(\zeta_{\mathbf{m}})} \sum_{|\mathbf{n}| \leq n} \left| \widetilde{f}_{\mathbf{m},\mathbf{n}}(\mathbf{x}) - \zeta_{\mathbf{m}} \left( \mathbf{x} - \frac{\mathbf{m}}{N} \right)^{\mathbf{n}} \right|$$

$$\leq \lambda_i \mu_i 2^{d+n+1} d^n (d+n) \epsilon.$$

We choose $\epsilon = \frac{\delta}{\lambda_i \mu_i 2^{d+n+2} d^n (d+n)}$, so that $\|\bar{f}_i - \widetilde{f}_i\|_\infty \leq \frac{\delta}{2}$. Thus, we eventually have $\|\widetilde{f}_i - f_i^\phi\|_\infty \leq \delta$. Now we compute the depth and computational units for implement $\widetilde{f}_i$. $\widetilde{f}_i$ can be implemented by a collection of parallel sub-networks that compute each $\widetilde{f}_{\mathbf{m},\mathbf{n}}$. The total number of parallel sub-networks is bounded by $d^n (N+1)^d$. For each sub-network, we observe that $\psi$ can be exactly implemented by a single layer ReLU network, i.e., $\psi(x) = \mathrm{ReLU}(x+2) - \mathrm{ReLU}(x+1) - \mathrm{ReLU}(x-1) + \mathrm{ReLU}(x-2)$. Corollary 1 shows that $\widehat{\times}_\epsilon$ can be implemented by a depth $c_1 \log \frac{1}{\epsilon}$ ReLU network. Therefore, the whole network for implementing $\widetilde{f}_i$ has no more than $c_1' \left( \log \frac{1}{\epsilon} + 1 \right)$ layers and $c_1' d^n (N+1)^d \left( \log \frac{1}{\epsilon} + 1 \right)$ neurons and weight parameters. With $\epsilon = \frac{\delta}{\lambda_i \mu_i 2^{d+n+2} d^n (d+n)}$ and $N = \left\lceil \left( \frac{\mu_i \lambda_i (2r)^{1-\alpha} 2^{d+n+2} d^{n+\alpha/2}}{\delta n!} \right)^{\frac{1}{n+\alpha}} \right\rceil$, we obtain that the whole network has no more than $c_1 \log \frac{1}{\delta}$ layers, and at most $c_2 \delta^{-\frac{d}{n+\alpha}} \left( \log \frac{1}{\delta} + 1 \right)$ neurons and weight parameters, for constants $c_1, c_2$ depending on $d, n$, and $f_i \circ \phi_i^{-1}$. $\qquad \square$

## B.3 Proof of Theorem 4

*Proof.* We expand the estimation error as

$$\left\| \widehat{f} - f \right\|_\infty = \left\| \sum_{i=1}^{C_{\mathcal{M}}} \widehat{\times}(\widehat{f}_i, \widehat{\mathbb{1}}_\Delta \circ \widehat{d}_i^2) - f \right\|_\infty$$

$$= \left\| \sum_{i=1}^{C_{\mathcal{M}}} \widehat{\times}(\widehat{f}_i, \widehat{\mathbb{1}}_\Delta \circ \widehat{d}_i^2) - f \rho_i \mathbb{1}(\mathbf{x} \in U_i) \right\|_\infty$$

$$\leq \sum_{i=1}^{C_{\mathcal{M}}} \left\| \widehat{\times}(\widehat{f}_i, \widehat{\mathbb{1}}_\Delta \circ \widehat{d}_i^2) - f_i \mathbb{1}(\mathbf{x} \in U_i) \right\|_\infty$$

$$\leq \sum_{i=1}^{C_{\mathcal{M}}} \left\| \widehat{\times}(\widehat{f}_i, \widehat{\mathbb{1}}_\Delta \circ \widehat{d}_i^2) - \widehat{f}_i \times (\widehat{\mathbb{1}}_\Delta \circ \widehat{d}_i^2) + \widehat{f}_i \times (\widehat{\mathbb{1}}_\Delta \circ \widehat{d}_i^2) - f_i \times (\widehat{\mathbb{1}}_\Delta \circ \widehat{d}_i^2) \right.$$
$$\left. + f_i \times (\widehat{\mathbb{1}}_\Delta \circ \widehat{d}_i^2) - f_i \times \mathbb{1}(\mathbf{x} \in U_i) \right\|_\infty$$

$$\leq \sum_{i=1}^{C_{\mathcal{M}}} \underbrace{\left\| \widehat{\times}(\widehat{f}_i, \widehat{\mathbb{1}}_\Delta \circ \widehat{d}_i^2) - \widehat{f}_i \times (\widehat{\mathbb{1}}_\Delta \circ \widehat{d}_i^2) \right\|_\infty}_{A_{i,1}} + \underbrace{\left\| \widehat{f}_i \times (\widehat{\mathbb{1}}_\Delta \circ \widehat{d}_i^2) - f_i \times (\widehat{\mathbb{1}}_\Delta \circ \widehat{d}_i^2) \right\|_\infty}_{A_{i,2}}$$

$$+ \underbrace{\left\| f_i \times (\widehat{\mathbb{1}}_\Delta \circ \widehat{d}_i^2) - f_i \times \mathbb{1}(\mathbf{x} \in U_i) \right\|_\infty}_{A_{i,3}}.$$

The first two terms $A_{i,1}, A_{i,2}$ are straightforward to handle, since by the construction we have

$$A_{i,1} = \left\| \widehat{\times}(\widehat{f}_i, \widehat{\mathbb{1}}_\Delta \circ \widehat{d}_i^2) - \widehat{f}_i \times (\widehat{\mathbb{1}}_\Delta \circ \widehat{d}_i^2) \right\|_\infty \leq \eta, \quad \text{and}$$

$$A_{i,2} = \left\| \widehat{f}_i \times (\widehat{\mathbb{1}}_\Delta \circ \widehat{d}_i^2) - f_i \times (\widehat{\mathbb{1}}_\Delta \circ \widehat{d}_i^2) \right\|_\infty \leq \left\| \widehat{f}_i - f_i \right\|_\infty \left\| \widehat{\mathbb{1}}_\Delta \circ \widehat{d}_i^2 \right\|_\infty \leq \delta.$$

By Lemma 3, we have $\max_{\mathbf{x} \in \mathcal{K}_i} |f_i(\mathbf{x})| \leq \frac{c(\pi+1)}{r(1-r/\tau)} \Delta$ for a constant $c$ depending on $f_i$. Then we bound $A_{i,3}$ as

$$A_{i,3} = \left\| f_i \times (\widehat{\mathbb{1}}_\Delta \circ \widehat{d_i^2}) - f_i \times \mathbb{1}(\mathbf{x} \in U_i) \right\|_\infty \leq \max_{\mathbf{x} \in \mathcal{K}_i} |f_i(\mathbf{x})| \leq \frac{c(\pi+1)}{r(1-r/\tau)} \Delta.$$

$\square$

## B.4   Proof of Lemma 3

*Proof.* We extend $f_i \circ \phi_i^{-1}$ to the whole cube $[0,1]^d$ as in the proof of Theorem 3. We also have $f_i(\mathbf{x}) = 0$ for $\|\mathbf{x} - \mathbf{c}_i\|_2 = r$. By the first order Taylor expansion, we have for any $\mathbf{x}, \mathbf{y} \in U_i$

$$
\begin{aligned}
|f_i(\mathbf{x}) - f_i(\mathbf{y})| &= \left| f_i \circ \phi_i^{-1}(\phi_i(\mathbf{x})) - f_i \circ \phi_i^{-1}(\phi_i(\mathbf{y})) \right| \\
&\leq \left\| \nabla(f_i \circ \phi_i^{-1})(\mathbf{z}) \right\|_2 \|\phi_i(\mathbf{x}) - \phi_i(\mathbf{y})\|_2 \\
&\leq \left\| \nabla(f_i \circ \phi_i^{-1})(\mathbf{z}) \right\|_2 b_i \|V_i\|_2 \|\mathbf{x} - \mathbf{y}\|_2,
\end{aligned}
$$

where $\mathbf{z}$ is a linear interpolation of $\phi_i(\mathbf{x})$ and $\phi_i(\mathbf{y})$ satisfying the mean value theorem. Since $f_i \circ \phi_i^{-1}$ is $C^n$ in $[0,1]^d$, the first derivative is uniformly bounded, i.e., $\left\| \nabla f_i \circ \phi_i^{-1}(\mathbf{z}) \right\|_2 \leq \alpha_i$ for any $\mathbf{z} \in [0,1]^d$. Let $\mathbf{y} \in U_i$ satisfying $f_i(\mathbf{y}) = 0$. In order to bound the function value for any $\mathbf{x} \in \mathcal{K}_i$, we only need to bound the Euclidean distance between $\mathbf{x}$ and $\mathbf{y}$. More specifically, for any $\mathbf{x} \in \mathcal{K}_i$, we need to show that there exists $\mathbf{y} \in U_i$ satisfying $f_i(\mathbf{y}) = 0$, such that $\|\mathbf{x} - \mathbf{y}\|_2$ is sufficiently small.

Before continuing with the proof, we introduce some notations. Let $\gamma(t)$ be a geodesic on $\mathcal{M}$ parameterized by the curve length. In the following context, we use $\dot\gamma$ and $\ddot\gamma$ to denote the first and second derivatives of $\gamma$ with respect to $t$. By the definition of geodesic, we have $\|\dot\gamma(t)\|_2 = 1$ (unit speed) and $\ddot\gamma(t) \perp \dot\gamma(t)$.

Without loss of generality, we shift $\mathbf{c}_i$ to $\mathbf{0}$. We consider a geodesic starting from $\mathbf{x}$ with initial "velocity" $\dot\gamma(0) = \mathbf{v}$ in the tangent space of $\mathcal{M}$ at $\mathbf{x}$. To utilize polar coordinate, we define two auxiliary quantities: $\ell(t) = \|\gamma(t)\|_2$ and $\theta(t) = \arccos \frac{\gamma(t)^\top \dot\gamma(t)}{\|\gamma(t)\|_2} \in [0, \pi]$. As can be seen in Figure 5, $\ell$ and $\theta$ have clear geometrical interpretations: $\ell$ is the radial distance from the center $\mathbf{c}_i$, and $\theta$ is the angle between the velocity and $\gamma(t)$.

Figure 7: Illustration of $\ell$ and $\theta$ along a parametric curve $\gamma$.

Suppose $\mathbf{y} = \gamma(T)$, we need to upper bound $T$. Note that $\ell(T) - \ell(0) \leq r - \sqrt{r^2 - \Delta} \leq \Delta/r$. Moreover, observe that the derivative of $\ell$ is $\dot\ell(t) = \cos\theta(t)$, since $\gamma$ has unit speed. It suffices to find a lower bound on $\dot\ell(t) = \cos\theta(t)$ so that $T \leq \frac{\Delta}{r \inf_t \dot\ell(t)}$.

We immediately have the second derivative of $\ell$ as $\ddot{\ell}(t) = -\sin\theta(t)\dot{\theta}(t)$. Meanwhile, using the equation $\ell(t) = \sqrt{\gamma(t)^\top\gamma(t)}$, we also have

$$\ddot{\ell}(t) = \frac{\left(\ddot{\gamma}(t)^\top\gamma(t) + \dot{\gamma}(t)^\top\dot{\gamma}(t)\right)\sqrt{\gamma(t)^\top\gamma(t)} - \left(\gamma(t)^\top\dot{\gamma}(t)\right)^2/\sqrt{\gamma(t)^\top\gamma(t)}}{\gamma(t)^\top\gamma(t)}. \tag{6}$$

Note that by definition, we have $\dot{\gamma}(t)^\top\dot{\gamma}(t) = 1$ and $\gamma(t)^\top\dot{\gamma}(t) = \cos\theta(t)\sqrt{\gamma(t)^\top\gamma(t)}$. Plugging into (6), we can derive

$$\ddot{\ell}(t) = \frac{1 + \ddot{\gamma}(t)^\top\gamma(t) - \cos^2\theta(t)}{\ell(t)} = \frac{\sin^2\theta(t) + \ddot{\gamma}(t)^\top\gamma(t)}{\ell(t)}. \tag{7}$$

Now we find a lower bound on $\ddot{\gamma}(t)^\top\gamma(t)$. Specifically, by Cauchy-Schwarz inequality, we have

$$\ddot{\gamma}(t)^\top\gamma(t) \geq -\|\ddot{\gamma}(t)\|_2 \|\gamma(t)\|_2 |\cos\angle(\ddot{\gamma}(t), \gamma(t))|$$
$$\geq -\frac{r}{\tau}|\cos\angle(\ddot{\gamma}(t), \gamma(t))|.$$

The last inequality follows from $\|\ddot{\gamma}(t)\|_2 \leq \frac{1}{\tau}$ (Niyogi et al., 2008) and $\|\gamma(t)\|_2 \leq r$. We now need to bound $\angle(\ddot{\gamma}(t), \gamma(t))$, given $\angle(\gamma(t), \dot{\gamma}(t)) = \theta(t)$ and $\ddot{\gamma}(t) \perp \dot{\gamma}(t)$. Consider the following optimization problem,

$$\min \quad a^\top x, \tag{8}$$
$$\text{subject to} \quad x^\top x = 1,$$
$$b^\top x = 0.$$

By assigning $a = \frac{\gamma(t)}{\|\gamma(t)\|_2}$ and $b = \frac{\dot{\gamma}(t)}{\|\dot{\gamma}(t)\|_2}$, the optimal objective value is exactly the minimum of $\cos\angle(\ddot{\gamma}(t), \gamma)$. Additionally, we can find the maximum of $\cos\angle(\ddot{\gamma}(t), \gamma)$ by replacing the minimization in (8) by maximization. We solve (8) by the Lagrangian method. More precisely, let

$$\mathcal{L}(x, \lambda, \mu) = -a^\top x + \lambda(x^\top x - 1) + \mu(b^\top x).$$

We have the optimal solution $x^*$ satisfying $\nabla_x\mathcal{L} = 0$, which implies $x^* = \frac{1}{2\lambda^*}(a - \mu^*b)$ with $\mu^*$ and $\lambda^*$ being the optimal dual variable. By the primal feasibility, we have $\mu^* = a^\top b$ and $\lambda^* = -\frac{1}{2}\sqrt{1 - (a^\top b)^2}$. Therefore, the optimal objective value is $-\sqrt{1 - (a^\top b)^2}$. Similarly, the maximum is $\sqrt{1 - (a^\top b)^2}$. Note that $a^\top b = \cos\theta(t)$, we then get

$$\ddot{\gamma}(t)^\top\gamma(t) \geq -\frac{r}{\tau}\sin\theta(t).$$

Substituting into (7), we have the following lower bound

$$\ddot{\ell}(t) = \frac{\sin\theta^2(t) + \ddot{\gamma}(t)^\top\gamma(t)}{\ell(t)} \geq \frac{1}{\ell(t)}\left(\sin^2\theta(t) - \frac{r}{\tau}\sin\theta(t)\right).$$

Now combining with $\ddot{\ell}(t) = -\sin\theta(t)\dot{\theta}(t)$, we can derive

$$\dot{\theta}(t) \leq -\frac{1}{\ell(t)}\left(\sin\theta(t) - \frac{r}{\tau}\right). \tag{9}$$

Inequality (9) has an important implication: When $\sin\theta(t) > \frac{r}{\tau}$, as $t$ increasing, $\theta(t)$ is monotone decreasing until $\sin\theta(t') = \frac{r}{\tau}$ for some $t' = t$. Thus, we distinguish two cases depending on the value of $\theta(0)$. Indeed, we only need to consider $\theta(0) \in [0, \pi/2]$. The reason behind is that if $\theta(0) \in (\pi/2, \pi]$, we only need to set the initial velocity in the opposite direction.

**Case 1**: $\theta(0) \in \left[0, \arcsin\frac{r}{\tau}\right]$. We claim that $\theta(t) \in \left[0, \arcsin\frac{r}{\tau}\right]$ for all $t \leq T$. In fact, suppose there exists some $t_1 \leq T$ such that $\theta(t_1) > \arcsin\frac{r}{\tau}$. By the continuity of $\theta$, there exists $t_0 < t_1$, such that $\theta(t_0) = \arcsin\frac{r}{\tau}$ and $\theta(t) \geq \arcsin\frac{r}{\tau}$ for $t \in [t_0, t_1]$. This already gives us a contradiction:

$$\theta(t_0) < \theta(t_1) = \theta(t_0) + \underbrace{\int_{t_0}^{t_1}\dot{\theta}(t)dt}_{\leq 0} \leq \theta(t_0).$$

Therefore, we have $\dot{\ell}(t) \geq \cos\arcsin\frac{r}{\tau} = \sqrt{1 - \frac{r^2}{\tau^2}}$, and thus $T \leq \frac{\Delta}{r\sqrt{1-\frac{r^2}{\tau^2}}}$.

**Case 2**: $\theta(0) \in \left(\arcsin\frac{r}{\tau}, \pi/2\right]$. It is enough to show that $\theta(0)$ can be bounded sufficiently away from $\pi/2$. Let $\gamma_{\mathbf{c},\mathbf{x}} \subset \mathcal{M}$ be a geodesic from $\mathbf{c}_i$ to $\mathbf{x}$. We analogously define $\theta_{\mathbf{c},\mathbf{x}}$ and $\ell_{\mathbf{c},\mathbf{x}}$ as for the geodesic from $\mathbf{x}$ to $\mathbf{y}$. Let $T_{r/2} = \sup\{t : \ell_{\mathbf{c},\mathbf{x}}(t) \leq r/2 - \Delta/r\}$, and denote $\mathbf{z} = \gamma_{\mathbf{c},\mathbf{x}}(T_{r/2})$. We must have $\theta_{\mathbf{c},\mathbf{x}}(T_{r/2}) \in [0, \pi/2]$ and $\ell_{\mathbf{c},\mathbf{x}}(T_{r/2}) = r/2 - \Delta/r$, otherwise there exists $T'_{r/2} > T_{r/2}$ satisfying $\ell_{\mathbf{c},\mathbf{x}}(T'_{r/2}) \leq r/2$. Denote $T_{\mathbf{x}}$ satisfying $\mathbf{x} = \gamma_{\mathbf{c},\mathbf{x}}(T_{\mathbf{x}})$. We bound $\theta_{\mathbf{c},\mathbf{x}}(T_{\mathbf{x}})$ as follows,

$$\theta_{\mathbf{c},\mathbf{x}}(T_{\mathbf{x}}) = \theta_{\mathbf{c},\mathbf{x}}(T_{r/2}) + \int_{T_{r/2}}^{T_{\mathbf{x}}} \dot{\theta}_{\mathbf{c},\mathbf{x}}(t) dt$$

$$\leq \frac{\pi}{2} - \int_{T_{r/2}}^{T_{\mathbf{x}}} \frac{1}{\ell_{\mathbf{c},\mathbf{x}}(t)} \left(\sin\theta_{\mathbf{c},\mathbf{x}}(t) - \frac{r}{\tau}\right) dt.$$

If there exists some $t \in (T_{r/2}, T_{\mathbf{x}}]$ such that $\sin\theta_{\mathbf{c},\mathbf{x}}(t) \leq \frac{r}{\tau}$, by the previous reasoning, we have $\sin\theta_{\mathbf{c},\mathbf{x}}(T_{\mathbf{x}}) \leq \frac{r}{\tau}$. Thus, we only need to handle the case when $\sin\theta_{\mathbf{c},\mathbf{x}}(t) > \frac{r}{\tau}$ for all $t \in (T_{r/2}, T_{\mathbf{x}}]$. In this case, $\theta_{\mathbf{c},\mathbf{x}}(t)$ is monotone decreasing, hence we further have

$$\theta_{\mathbf{c},\mathbf{x}}(T_{\mathbf{x}}) \leq \frac{\pi}{2} - \int_{T_{r/2}}^{T_{\mathbf{x}}} \frac{1}{\ell_{\mathbf{c},\mathbf{x}}(t)} \left(\sin\theta_{\mathbf{c},\mathbf{x}}(T_{\mathbf{x}}) - \frac{r}{\tau}\right) dt$$

$$\leq \frac{\pi}{2} - (T_{\mathbf{x}} - T_{r/2})\frac{1}{r} \left(\sin\theta_{\mathbf{c},\mathbf{x}}(T_{\mathbf{x}}) - \frac{r}{\tau}\right)$$

$$\leq \frac{\pi}{2} - \frac{1}{2} \left(\sin\theta_{\mathbf{c},\mathbf{x}}(T_{\mathbf{x}}) - \frac{r}{\tau}\right).$$

The last inequality follows from $T_{\mathbf{x}} - T_{r/2} \geq r/2$. Using the fact, $\sin x \geq \frac{2}{\pi}x$, we can derive

$$\theta_{\mathbf{c},\mathbf{x}}(T_{\mathbf{x}}) \leq \frac{\pi}{2} - \frac{1}{2} \left(\frac{2}{\pi}\theta_{\mathbf{c},\mathbf{x}}(T_{\mathbf{x}}) - \frac{r}{\tau}\right)$$

$$\implies \theta_{\mathbf{c},\mathbf{x}}(T_{\mathbf{x}}) \leq \frac{\pi}{2} \left(\frac{\pi + r/\tau}{\pi + 1}\right).$$

We can then set $\theta(0) = \theta_{\mathbf{c},\mathbf{x}}(T_{\mathbf{x}})$, and thus

$$\cos\theta(0) \geq \cos\left(\frac{\pi}{2}\frac{\pi + r/\tau}{\pi + 1}\right) = \cos\left(\frac{\pi}{2}\left(1 - \frac{1 - r/\tau}{\pi + 1}\right)\right)$$

$$= \sin\left(\frac{\pi}{2}\frac{1 - r/\tau}{\pi + 1}\right)$$

$$\geq \frac{1 - r/\tau}{\pi + 1}.$$

Therefore, we have $T \leq \frac{\Delta}{r\cos\theta(0)} \leq \frac{\pi+1}{r(1-r/\tau)}\Delta$. By the choice of $r < \tau/2$, we immediately have $\frac{\tau}{\sqrt{\tau^2-r^2}} < \frac{\pi+1}{1-r/\tau}$. Hence, combining case 1 and case 2, we conclude

$$T \leq \frac{\pi + 1}{r(1 - r/\tau)}\Delta.$$

Therefore, the function value $f(\mathbf{x})$ on $\mathcal{K}_i$ is bounded by $\alpha_i\frac{\pi+1}{r(1-r/\tau)}\Delta$. It suffices to let $c = \max_i \alpha_i b_i \|V_i\|_2$, and we complete the proof. $\square$

## B.5 Characterization of the Size of the ReLU Network

*Proof.* We evenly split the error $\epsilon$ into 3 parts for $A_{i,1}, A_{i,2}$, and $A_{i,3}$, respectively. We pick $\eta = \frac{\epsilon}{3C_{\mathcal{M}}}$ so that $\sum_{i=1}^{C_{\mathcal{M}}} A_{i,1} \leq \frac{\epsilon}{3}$. The same argument yields $\delta = \frac{\epsilon}{3C_{\mathcal{M}}}$. Analogously, we can choose $\Delta = \frac{r(1-r/\tau)\epsilon}{3c(\pi+1)C_{\mathcal{M}}}$. Finally, we pick $\nu = \frac{\Delta}{16B^2D}$ so that $8B^2D\nu < \Delta$.

Now we compute the number of layers, and the number of neurons and weight parameters in the ReLU network identified by Theorem 1.

1. For the chart determination sub-network, $\widehat{\mathbb{1}}_\Delta$ can be implemented by a single layer ReLU network. The approximation of the distance function $\widehat{d_i^2}$ can be implemented by a network of depth $O\left(\log \frac{1}{\nu}\right)$ and the number of neurons and weight parameters is at most $O\left(\log \frac{1}{\nu}\right)$. Plugging in our choice of $\nu$, we have the depth is no greater than $c_1\left(\log \frac{1}{\epsilon} + \log D\right)$ with $c_1$ depending on $d, f, \tau$, and the surface area of $\mathcal{M}$. The number of neurons and weight parameters is also $c_1'\left(\log \frac{1}{\epsilon} + \log D\right)$ except for a different constant. Note that there are $D$ parallel networks computing $\widehat{d_i^2}$ for $i = 1, \ldots, C_\mathcal{M}$. Hence, the total number of neurons and weight parameters is $c_1' C_\mathcal{M} D \left(\log \frac{1}{\epsilon} + \log D\right)$ with $c_1'$ depending on $d, f, \tau$, and the surface area of $\mathcal{M}$.

2. For the Taylor polynomial sub-network, $\phi_i$ can be implemented by a linear network with at most $Dd$ weight parameters. To implement each $\widehat{f_i}$, we need a ReLU network of depth $c_4 \log \frac{1}{\delta}$. The number of neurons and weight parameters is $c_4' \delta^{-\frac{d}{n+\alpha}} \log \frac{1}{\delta}$. Here $c_4, c_4'$ depend on $n, d, f_i \circ \phi_i^{-1}$. Substituting $\delta = \frac{\epsilon}{3C_\mathcal{M}}$, we get the depth is $c_2 \log \frac{1}{\epsilon}$ and the number of neurons and weight parameters is $c_2' \epsilon^{-\frac{d}{n+\alpha}} \log \frac{1}{\epsilon}$. There are totally $C_\mathcal{M}$ parallel $\widehat{f_i}$'s, hence the total number of neurons and weight parameters is $c_2' C_\mathcal{M} \epsilon^{-\frac{d}{n+\alpha}} \log \frac{1}{\epsilon}$ with $c_2'$ depending on $d, n, f_i \circ \phi_i^{-1}, \tau$, and the surface area of $\mathcal{M}$.

3. For the product sub-network, the analysis is similar to the chart determination sub-network. The depth is $O\left(\log \frac{1}{\eta}\right)$, and the number of neurons and weight parameters is $O\left(\log \frac{1}{\eta}\right)$. The choice of $\eta$ yields the depth is $c_3 \log \frac{1}{\epsilon}$, and the number of neurons and weight parameters is $c_3' \log \frac{1}{\epsilon}$. There are $C_\mathcal{M}$ parallel pairs of outputs from the chart determination and the Taylor polynomial sub-networks. Hence, the total number of weight parameters is $c_3' C_\mathcal{M} \log \frac{1}{\epsilon}$ with $c_3'$ depending on $d, \tau$, and the surface area of $\mathcal{M}$.

Combining these 3 sub-networks, we see the depth of the full network is $c\left(\log \frac{1}{\epsilon} + \log D\right)$ for some constant $c$ depending on $d, n, f, \tau$, and the surface area of $\mathcal{M}$. The total number of neurons and weight parameters is $c'\left(\epsilon^{-\frac{d}{n+\alpha}} \log \frac{1}{\epsilon} + D \log \frac{1}{\epsilon} + D \log D\right)$ for some constant $c'$ depending on $d, n, f, \tau$, and the surface area of $\mathcal{M}$. $\qquad \square$