[Reviews · NeurIPS 2019]

Reviewer 1



The paper is written well and quite easy to follow. The proof idea is relatively straightforward (one can learn a subnetwork for each corresponding chart and an indicator to determine to determine which charts are "active"). The main idea, as in Yarotsky (2017) is to approximate the functions by a Taylor series. As Yarotsky showed, this can be done efficiently by neural network. However, there are still some technical challenges and the main technical contribution of the paper is to overcome these challenges. Edit: I have read the author response and other reviews. My opinion is still that this paper is well-motivated and is one direction to theoretically understand why neural networks are able to efficiently fit the data well. As another reviewer mentioned, the technical details are quite hard to follow so it would be great if this could be addressed (perhaps by making the body less dense). Nonetheless, for someone who is somewhat familiar with the arguments in Yarotsky, the bird's eye view of the argument is somewhat clear.

Reviewer 2



The paper proves that deep ReLU networks can approximate smooth functions defined on low dimensional compact manifolds where the number of neurons depends exponentially on the dimension of the manifold instead of the dimensional of the ambient space. Given a function f, the network consists of three main subnetworks: (1) a subnetwork that given a point returns the chart it belongs to (2) a subnetwork that approximates f on this chart and (3) a pairing subnetwork that combines the results of the first two networks. I like the motivation and the research question that is raised in this paper. The main motivation for considering this problem is the fact that many important data types (e.g. images, text) are believed to occupy only a low dimensional manifold in the ambient space. Previous universality results imply that the size of the network has to be exponential in D in order to get a specific approximation bound. Nevertheless, many networks with significantly fewer neurons perform very well. The theoretical result in this paper can be seen as a possible reason for their success since it shows that good approximation can be obtained by a network that whose size is exponential in d, which is often much smaller. The clarity of the paper varies. Section (1-3) are very well written and easy to follow. The informal version of the theorem on page 2 helps to understand the main result of the paper. The figures are also very helpful. On the other hand, I found the main technical part (section 4) very hard to follow. I was not able to verify all the mathematical details. A few concrete questions/comments: Step 3: there is an ambiguity in determining the chart to which a point belongs to, how this is solved? Step 4: Taylor approximation has local error guarantees in general, in contrast to the L infinity approximation used in this paper. Can the authors discuss this early in the paper? Information theoretic bounds - can the authors elaborate? This part was not clear as well. To conclude, I think that this paper targets an interesting question and provides a very nice result. My relatively low score is due to the fact that the technical section is very hard to read. I think that in its current form this paper might not be accessible to a lot of people in the community. ************************** Post Rebuttal: ************************** After reading the other reviews and the author response I stay positive and decided to raise my score to 7. I still think the paper presents an important theoretical contribution and I trust the authors to do the best they can in order to make it more accessible. In addition, from my perspective, the fact that there are no experiments is not problematic.

Reviewer 3



Originality: A new theoretical framework of approximating functions defined on low dimensional manifolds with ReLU networks Clarity: Needs more improvement (see below) Quality: Needs more improvement (see below) Significance: The topical area is important because ReLU networks are widely applied in deep learning More details comments: In the abstract, line 2, please be more specific when denoting line of research; this line seems very generic. Line 14, you say you implement a sub-network but there are no experimental results to demonstrate that, please reword with correct terminology. Line 48, it is not intuitive what it scales to, please consider rewriting it Line 68, you denote the \sigma function for sigmoidal then you use the notation again for the quadratic rectifier. Please stay consistent with terminology. Also line 68, max(0,x^2) = x^2, it looks like a typo since it will be non-negative. Line 145 (definition 6), to cite https://arxiv.org/pdf/1705.04565.pdf since they have similar definitions that the two books you cite do not contain Moreover, their definition of reach implies that \tau is ALWAYS zero because p can equal q. If we say that p \ne q, then we get something that makes sense. HOWEVER, we can still get \tau = 0, for example when M = {(x,x): x in R^+} \cup {(0,x): x in R^+}. Hence, their statement that \tau > 0 is false. Line 153 the claim you make with small \tau requiring multiple charts is only for your method. The M denoted above has \tau = 0 but only requires one chart. Line 159 (Assumption 1). Please define that B is an arbitrary constant. Line 160 (Assumption 2), \tau was already a definition. It is unclear why it would need to be an assumption. Line 177, it is not clear at all if it is possible to partition a manifold with open sets. Additionally, it is not clear if you need finite or infinite open sets, if it is even possible. Figure 2 there needs to be a caption with the definitions of the terms used in the figure. The terms are defined later on but we needed to scroll down several pages, so for completeness when looking at the figure, it would be better to define it. Line 208, Please define that the function you are trying to implement which is f(x,y) = xy Line 213, your subheadings are inconsistent in these sections, either capitalize the subsection headings or do not. Please set a precedent since in Step 1 you only capitalize the first word but in step 2 you capitalize. Thickness footnote needs to be elaborated in detail. Line 240, please consider renaming the approximate indicator function, it is a bit confusing the terminology since it is more of a unit step function. Line 269, do you mean uniform convergence? If so, please do write detail it. Line 272, is each term in the Taylor expansion a different layer of the network? That is what it seems like but by the wording it can be confusing. Line 324, please cite that relu networks are easier to train. Line 332, please cite again the papers that make the claim. Discussion, it would be better to have a paragraph that summarizes the conclusion instead of 5 sub-paragraphs that feel disjointed and not cohesive. You discuss implementing the network, but there are no experimental results. Since this paper is theoretical, it would benefit a lot if some experimental results were provided.

Reviewer 4



The paper is well written but the notions of atlas, chart, reach may be arcane to most readers here. I don't know what is the best way to simplify this but a short paragraph with simple informal descriptions of these terms would go a long way. The result is nice and also introduces several important concepts from Manifold theory, but the proof idea is pretty much as one would expect given the assumptions of bounded reach giving a bounded atlas and continuity of several order of derivatives allowing a taylor approximation within each chart. Of course any intuition about why any such functions (even with additional assumptions) may be learnable by gradient descent would be much more interesting as that is the bigger mystery of deep learning.

[Author Response · NeurIPS 2019]

# Author Response to Reviews on "Efficient Approximation of Deep ReLU Networks"

We appreciate all reviewers' valuable comments. Here are our response to the major questions raised by the reviewers.

**To Reviewer #1**:

**Q**: Line 174, regarding the approximation of $f$ by polynomials and construction of $\phi$.

**A**: We construct $\phi_i$'s as linear transformations (Line 222) so it can be realized by a simple linear network. To approximate $f$, we first decompose $f = \sum_{i=1}^{C_{\mathcal{M}}} f_i$ as in Line 248. Our Taylor approximation sub-network approximates each $f_i$ in two components: the first one realizes the linear projection $\phi_i$ by a linear network and the second one approximates $f_i \circ \phi_i^{-1}$ in the neighborhood $U_i$ by a ReLU network (Theorem 3). We will clarify the realization of $\phi_i$'s in the next version.

**To Reviewer #2**:

**Q**: Step 3: there is an ambiguity in determining the chart to which a point belongs to, how this is solved?

**A**: We allow a point $\mathbf{x}$ to belong to multiple charts, and the chart determination sub-network determines all the proper charts that $\mathbf{x}$ belongs to (Line 231). Specifically, the $U_i$'s form an open cover of $\mathcal{M}$ (Line 216). Thus, a given input $\mathbf{x}$ can belong to multiple $U_i$'s. For the approximation of $f$, we associate each $U_i$ with a weight $\rho_i(\mathbf{x})$ from a partition of unity satisfying $\sum_i \rho_i(\mathbf{x}) = 1$ for all $\mathbf{x} \in \mathcal{M}$. Then our Taylor approximation sub-network approximates $f_i(\mathbf{x}) = \rho_i(\mathbf{x})f(\mathbf{x})$ (Line 248). Consequently, the sum of all the outputs from the pairing sub-network (products of the indicator function of $U_i$ and the corresponding Taylor approximation for $f_i(\mathbf{x})$, Line 280) approximates $f(\mathbf{x}) = \sum_{i=1}^{C_{\mathcal{M}}} f_i(\mathbf{x})$.

**Q**: Taylor approximation has local error guarantees in general, in contrast to the $L_\infty$ approximation used in this paper.

**A**: While Taylor approximation yields a local error guarantee in each $U_i$, our $L_\infty$ error bound holds uniformly for $\mathbf{x} \in \mathcal{M}$. A uniform upper bound of all local errors gives rise to the $L_\infty$ error bound (Theorem 4). In this paper, we uniformly bound all local errors and therefore the result is given in the $L_\infty$ error.

**Q**: Information theoretic bounds - can the authors elaborate? Improve section 4, Ideally.

**A**: We show our obtained network size matches the lower bound up to log factors (Lines 191 - 193). We will rephrase "information-theoretic bound" as "the lower bound in Theorem 2". We will elaborate on this part in the revision.

We will also improve the technical Section 4 by including more high-level ideas and some graphical illustrations.

**To Reviewer #3**:

**Q**: The authors show no experimental results; instead they reference other networks (e.g., VGG, Alexnet, etc.).

**A**: There have been empirical evidences (VGG, Alexnet, etc.) revealing a huge gap between the network size used in practice and the one predicted by existing theories (Line 52). Therefore, we believe it is not necessary to provide our own experimental results. Our theoretical results bridge this gap by taking low dimensional data structures into consideration, and establish efficient approximation theories for ReLU networks.

**Q**: Line 14, you say you implement a sub-network but there are no experimental results.

**A**: "Implementation" here means "analytical construction", which is a standard notion in approximation theory literature. We will use "construct" in the next version to avoid confusion.

**Q**: Line 48, it is not intuitive what it scales to, please consider rewriting it.

**A**: We will rephrase it as "To achieve an $\epsilon$ uniform approximation error, the number of neurons scales as $\epsilon^{-256 \times 256 \times 3}$ ([Barron, 1993, Universal approximation bounds]). Setting $\epsilon = 0.1$ gives rise to $10^{256 \times 256 \times 3}$ neurons." in the revision.

**Q**: Line 145 (definition 6), to cite https://arxiv.org/pdf/1705.04565.pdf, and comments on the reach.

**A**: We will add more citations in the next version. Our definition on the reach (Definition 6) is consistent with that in [arXiv:1705.04565]: The set $\mathcal{C}(\mathcal{M})$ (Line 145) contains all points having two closest points in $\mathcal{M}$. Reach is defined as the minimum distance between $\mathcal{M}$ and $\mathcal{C}(\mathcal{M})$. The reach of $\mathcal{M} = \{(x, x) : x \in \mathbb{R}^+\} \cup \{(0, x) : x \in \mathbb{R}^+\}$ is 0, however, this is not a smooth manifold. Our paper considers Riemannian manifold (Line 122) and is therefore smooth. It is generally true that a smooth manifold with a small reach needs a large number of charts (open balls in Line 216).

**Q**: Line 159 the constant $B$, and Line 160 the assumption on reach.

**A**: We will rephrase Line 159 to "There exists $B > 0$ such that, for any $\mathbf{x} \in \mathcal{M}$, we have $|x_i| \le B$ for $i = 1, \ldots, D$."

Assumption 2 says that $\mathcal{M}$ has a positive reach. We will rephrase Assumption 2 to "The reach of $\mathcal{M}$ is $\tau > 0$."

**Q**: Line 177, it is not clear at all if it is possible to partition a manifold with open sets.

**A**: We assume the manifold $\mathcal{M}$ is compact (Assumption 1, Line 158). Hence, the existence of a finite open cover is guaranteed by Heine-Borel theorem (See Wikipedia and Folland, Real Analysis, 1999).

**Q**: Line 272, is each term in the Taylor expansion a different layer of the network?

**A**: As shown in the proof of Theorem 3 (Lines 499 - 504), each term $\widetilde{f}_{\mathbf{m},\mathbf{n}}$ in the Taylor expansion is approximated by a ReLU network of depth at most $c_1 \log \frac{1}{\delta}$. There are totally $d^n(N+1)^d$ terms in the Taylor expansion ($N$ is chosen as in Line 505). Therefore, the number of terms in the Taylor expansion essentially indicates the width of our Taylor approximation sub-network.

[Meta-Review · NeurIPS 2019]

The is an interesting result on a well-motivated problem of central interest. The paper is written well.